# Provably Efficient Learning in Partially Observable Contextual Bandit

## Abstract

In this paper, we investigate transfer learning in partially observable contextual bandits, where agents have limited knowledge from other agents and partial information about hidden confounders. We first convert the problem to identifying or partially identifying causal effects between actions and rewards through optimization problems. To solve these optimization problems, we sample compatible causal models via sequentially solving linear programs to obtain causal bounds with the consideration of estimation error. Our sampling algorithms provide desirable convergence results for suitable sampling distributions. We then show how causal bounds can be applied to improving classical bandit algorithms and affect the regrets with respect to the size of action sets and function spaces. Notably, in the task with function approximation which allows us to handle general context distributions, our method improves the order dependence on function space size compared with previous literatures. We formally prove that our causally enhanced algorithms outperform classical bandit algorithms and achieve orders of magnitude faster convergence rates. Finally, we perform simulations that demonstrate the efficiency of our strategy compared to the current state-of-the-art methods.

## 1 Introduction

Bandit learning involves decision-making to minimize regrets (Lattimore and Szepesvári, 2020), but classical bandit algorithms lack prior knowledge and can be computationally expensive. To address this, transfer learning techniques leverage knowledge from related tasks, relying on domain expertise and invariant knowledge (Zhuang et al., 2021). However, in practical scenarios like autonomous driving, human drivers have experiential information not captured by sensors, leading to transfer learning problems in partially observable contextual bandit settings (TLPOCB).

In TLPOCB, the expert has contextual information but transfers only partial information to the agent due to factors such as privileged information, privacy concerns, or physical limitations. However, using expert data without comprehending the expert context can result in biased learned policies that require a considerable amount of new data samples to improve. Causal inference has been widely employed in addressing unobserved contextual information in bandit and transfer learning problems, as discussed in (Bareinboim et al., 2015; Zhang and Bareinboim, 2021; Cai et al., 2022; Liu et al., 2021; Gong and Zhang, 2022). We outline three TLPOCB tasks and develop new causal bounds for discrete context and reward distributions. These bounds help learner agents achieve faster learning rates with limited source requirements by reducing exploration regrets in bandit algorithms.

Our contributions can be summarized as follows. Firstly, we introduce new tight causal bounds for partially observable transfer learning algorithms. To approximate these bounds, we develop sampling algorithms that sequentially solve refined linear programming problems to obtain valid samples. We demonstrate that under mild assumptions on the sampling distribution and the optimization procedure, the ordered statistics generated by our algorithms can almost surely converge to the solutions to the discrete optimization problems. Additionally, our algorithms incorporate estimation error, which is often neglected in previous literature (Zhang and Bareinboim, 2021; Li and Pearl, 2022; Liu et al., 2021). Secondly, we provide theoretical evidence that our algorithms consistently outperform classical algorithms, as shown in Table 1. Our theoretical results illustrate how our bounds impact the gap-dependent regrets and minimax regrets in terms of bandit quantities, such as the cardinality

of action sets. Moreover, our transfer learning algorithms only require several causal bounds rather than the whole dataset, which may show advantages when transferring the whole dataset is costly. Thirdly, we develop a context-dependent method to handle the bounds and improve the learning rate in transfer learning with function approximation. Our regret demonstrates an improvement in dependence on policy space $\Pi$ from $\sqrt{|\Pi|}$ to $\sqrt{\log |\Pi|}$, compared with (Zhang and Bareinboim, 2021). Finally, we conduct various numerical experiments to verify our findings.

## 1.1 RELATED WORK

Our work is related to prior research on causal bounds. In (Pearl, 2009), a general calculus known as do-calculus is developed for identifying causal effects using probabilistic tools. For non-identifiable causal problems, Tian and Pearl (2002) derives a model-free causal bound. Zhang and Bareinboim (2017) formulate a linear programming approach to obtain a causal bound in a discrete setting, while Shridharan and Iyengar (2022) makes this approach scalable. However, these LP-based methods may not be tight and rely on structural assumptions on hidden confounders. Li and Pearl (2022) is the first to consider partially observable back-door and front-door criteria and employs a non-linear optimization problem to compute causal bounds. Nonetheless, solving a non-linear optimization problem is computationally intensive and may get trapped in local optima. Duarte et al. (2023) summarize these optimization-based methods and provides an automated approach to causal inference in discrete settings. Zhang et al. (2022) employ a Markov Chain Monte Carlo (MCMC) method to approximate causal bounds using observational and experimental data. However, this method is not directly applicable in our setting because the agent in our scenario has access to partial knowledge about latent confounders, whereas Zhang et al. (2022) make structural assumptions on latent confounders. As a result, Zhang et al. (2022) only need sample a random vector from Dirichlet distributions while we need to sample constrained table from distributions supported on more complex simplex.

Our work is also closely linked to transfer learning. Zhang and Bareinboim (2017); Lazaric et al. (2013) investigate transfer learning in multi-armed bandit problems. Cai et al. (2022) examine transfer learning in contextual bandits with covariate shift, where two tasks share the same reward function (causal effects) across two bandits. Liu et al. (2018) apply transfer learning techniques in recommendation systems. Liu et al. (2021) extend the ideas in (Zhang and Bareinboim, 2017; 2021) to reinforcement learning settings. A similar work is (Tennenholtz et al., 2021), which uses a partially missing offline dataset to enhance the performance of online learning in linear bandits. Though several papers (Park and Faradonbeh; 2021) investigate partially observable contextual bandits, we found few papers focusing on transfer learning in this setting.

## 2 PRELIMINARIES

*Causal models.* In this section, we introduce the basic notations and definitions used throughout the paper. We use capital letters to denote variables ($X$) and small letters for their values ($x$). Let $\mathcal{X}$ stand for the domain of $X$, and $|\mathcal{X}|$ for its cardinality if $X$ is finite and discrete. We use $F(x)$ to represent the cumulative distribution function for random variable $X$.

We will use structural causal models (SCMs) (Powell, 2018) as the basic semantical framework of our analysis. A SCM $\mathcal{M}$ consists of a set of observed (endogenous) variables and unobserved (exogenous) variables. The values of each endogenous variable are determined by a structural function taking as argument a combination of the other endogenous and exogenous variables. A causal diagram associated with the SCM $\mathcal{M}$ is a directed acyclic graph where nodes correspond to variables and edges represent causal relations (see the causal graph of TLPOCB in Figure 1).

*Multi-armed bandit.* We now define the MAB setting using causal language. An agent for a stochastic MAB is given a SCM $\mathcal{M}$ with a decision node $A$ representing the arm selection and an outcome variable $Y$ representing the reward. For arm $a \in \mathcal{A}$, its expected reward $\mu_a$ is thus the effect of the action $do(a)$, i.e., $\mu_a = \mathbb{E}[Y|do(a)]$. Let $\mu^* = \max_{a \in \mathcal{A}} \mu_a$ and $a^*$ denote the optimal expected reward and the optimal arm, respectively. At each round $t = 1, 2, \cdots, T$, the agent performs an action $do(A_t = a_t)$ and observes a reward $Y_t$. The objective of the agent is to minimize the cumulative regret, namely, $Reg(T) = T\mu^* - \sum_{t=1}^{T} \mathbb{E}[Y_t]$.

*Contextual bandit.* We then describe the contextual bandit in causal setting. An agent for a contextual bandit is given a SCM $\mathcal{M}$ with an observed context variable $W$ besides the arm selection node $A$ and a reward variable $Y$. At each round $t = 1, 2, \cdots, T$, the agent can observe a context $w_t$ and performs an action $do(A_t = a_t)$ based on contexts and historical information. For each arm $a \in \mathcal{A}$, its expected reward given the context $w$ is defined as $\mu_{a,w} = \mathbb{E}[Y|do(a), w]$. Let $\mu_w^*$ denote the optimal expected reward with respect to the context $w$. The objective of the agent is to minimize the cumulative regret, namely, $Reg(T) = \sum_{t=1}^{T} \mathbb{E}[\mu_{w_t}^* - \mu_{a,w_t}|w_t]$.

However, the regret in such setting typically scales with $|\mathcal{W}|$. A practical way to avoid dependent on $|\mathcal{W}|$ is to use function approximation methods. We assume that the agent has access to a class of reward functions $\mathcal{F} \subset \mathcal{W} \times \mathcal{A} \to [0, 1]$ (e.g., linear function classes) that characterizes the mean of the reward distribution for a given context-action pair.

**Assumption 2.1** *There exists a function $f^* \in \mathcal{F}$ such that $f(w, a) = E[Y|w, do(a)]$.*

This *realizability assumption* is standard in contextual bandit literature (Simchi-Levi and Xu, 2021; Foster and Rakhlin, 2020; Foster et al., 2020). Let $\pi_f(w) = \arg\max_{a \in \mathcal{A}} f(w, a)$ denote the policy induced by the regression function $f \in \mathcal{F}$. The set of all induced policies forms the policy space $\Pi = \{\pi_f | f \in \mathcal{F}\}$. The objective of the agent is to achieve low regret with respect to the optimal policy $\pi_{f^*}$, and the cumulative regret over time horizon $T$ is defined as: $Reg(T) = \sum_{t=1}^{T} \mathbb{E}[f^*(w_t, \pi_{f^*}(w_t)) - f^*(w_t, \pi_f(w_t))|w_t]$.

The presence of unobserved confounder $U$ can make the transfer learning tasks challenging. Next, we give an example to show that direct approaches can lead to a policy function that is far from the true optimal policy.

**Example.** Consider a 2-arm contextual bandit problem, where the action $A$, context $U, W$ are binary. At the beginning of each round, two binary contextual variables $U, W$ are sampled from two independent Bernoulli distribution, $\mathbb{P}(U = 1) = 0.9, \mathbb{P}(W = 1) = 0.5$. The context variables $U, W$ affect the reward $Y$ received during each round according to Table 3 in Appendix A.

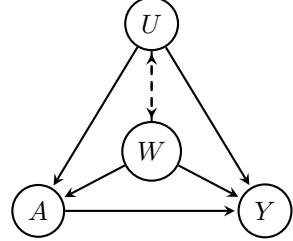

Fig. 1: A causal model with unobserved context $U$ and observed context $W$, where the causal relationship between $W$ and $U$ can be arbitrary.

The expert agent can observe the contextual variable $U, W$ and executes a context-aware optimal policy $\pi_{expert}^* : \mathcal{W} \times \mathcal{U} \to \mathcal{A}$. As a result, the expert generates multiple data points to obtain an observational distribution $\mathbb{P}(A, W)$ as shown in Table 4. Now we compute the conditional expectation $\mathbb{E}[Y|A = 0, W] = 1.81$, and $\mathbb{E}[Y|A = 1, W] = 0.9, \forall W \in \{0, 1\}$. Therefore, the optimal policy is $\pi_{agent}^*(W) = 0$ with probability 1. However, the policy differs a lot from $\pi_{expert}^*$ as $\pi_{expert}^*$ mainly selects the arm 1.

Upon closer examination of the example, it becomes clear that the asymmetrical probability mass function of the hidden context variable $U$ leads to a significant difference between the optimal expert and agent policies. Specifically, for the context $U = 1$ that occurs frequently, the rewards of the two arms are relatively similar, while for the rarely occurring context $U = 0$, the rewards differ significantly between the two arms. If the agent has knowledge of the distribution of $U$, it can infer that the observed rewards may be heavily influenced by the skewness from $F(u)$ and adjust its decision-making accordingly to make the correct choices.

## 3 TRANSFER LEARNING VIA CAUSAL INFERENCE

We consider an off-policy learning problem between an contextual bandit expert and an agent. The expert summarizes its experiences as the joint distribution $\hat{F}(a, y, w)$. However, there exists a latent confounder $U$ that affects the causal effects between actions and rewards, making the observed contextual bandit model incomplete. The agent wants to be more efficient and reuse the observed $\hat{F}(a, y, w)$ along with the extra prior knowledge about $U$, i.e., $\hat{F}(u)$, to find the optimal policy faster. Although $U$ is unobserved during online learning, its distribution can be usually known regardless of the model itself (e.g., $U$ stands for gender, gene type, blood type, or age). This transfer scenario is

depicted in Figure 1, where the actions $A$, outcomes $Y$, context $W$ and $U$, and causal structures used by the expert and the agent are identical.

### 3.1 TRANSFER LEARNING IN MULTI-ARMED BANDIT

In the first task, the MAB agent is unable to observe contexts $W$ and $U$, so its objective is to minimize the MAB regret. We will further examine the case where the agent can partially observe contexts.

**Causal bounds.** One can find the optimal action $do(A = a^*)$ by evaluating $\mathbb{E}[Y|do(a)]$. Therefore, the off-policy learning problem is equivalent to identifying the causal effect $\mathbb{E}[Y|do(a)]$ given the observational distribution $F(a, y, w)$ and prior knowledge $F(u)$. However, the causal effects of the action $A$ on the reward $Y$ is non-identifiable from incomplete experiences $F(a, y, w)$ and $F(u)$. This means that the causal effects of $A$ on $Y$ cannot be estimated without bias, regardless of sample size.

However, we can obtain tight bounds on the causal effects via an optimization problem when given a prior distribution $F(u)$ and $F(a, y, w)$. The optimization problem is as follows:

$$
\begin{aligned}
&\sup / \inf \mathbb{E}_{\mathcal{M}}[Y|do(a_0)] \\
s.t. \quad &F_{\mathcal{M}}(a, y, w) = F(a, y, w), \\
&F_{\mathcal{M}}(u) = F(u).
\end{aligned}
\tag{1}
$$

Here, the sup/inf is taken over all causal models $\mathcal{M}$, and $F_{\mathcal{M}}$ represents the distributions in the causal model $\mathcal{M}$. We can obtain causal upper and lower bounds by solving corresponding optimization problems. Since the optimization is performed over all compatible causal models, the resulting bounds are guaranteed to be tight. The specific formulation of (1) is presented in Theorem A.1. Our theorem generalizes that of (Li and Pearl, 2022) to random variables with estimation error, while Li and Pearl (2022) only consider the case of discrete random variables without estimation error.

The original optimization problem is challenging to solve but can be naturally simplified by probability mass functions for discrete random variables with bounded support. To extend this approach to general random variables, we discretize $\mathcal{A}, \mathcal{W}, \mathcal{U}$ and $\mathcal{Y}$ into $n_{\mathcal{A}}, n_{\mathcal{W}}, n_{\mathcal{U}}$ and $n_{\mathcal{Y}}$ disjoint blocks, respectively. For random variables that take in finite values, they have natural discretization. We first define $x_{ijkl} = \int_{a \in \mathcal{A}_i, y \in \mathcal{Y}_j, w \in \mathcal{W}_k, u \in \mathcal{U}_l} dF(a, y, w, u)$ and then integrate both sides of the first equality over $\mathcal{A}_i, \mathcal{Y}_j$, and $\mathcal{W}_k$ to obtain $\sum_l x_{ijkl} = \int_{a \in \mathcal{A}_i, y \in \mathcal{Y}_j, w \in \mathcal{W}_k} dF(a, y, w)$. This allows us to reformulate the first equality in terms of integration instead of the original pointwise equality. For the second constraint, we integrate this constraint using the same approach and get $\sum_{ijk} x_{ijkl} = \int_{u \in \mathcal{U}_l} dF(u)$. Compared with the original constraints in distributions, the linear constraints for $x_{ijkl}$ are much easier to handle. After discretization, the objective of the original optimization problem is $\hat{\mathbb{E}}_{\mathcal{M}}[Y|do(a_i)] = \sum_{j,k,l} \frac{y_j x_{ijkl} \sum_{i',j'} x_{i'j'kl}}{\sum_{j'} x_{ij'kl}}$, where $y_j$ is chosen to be $\int_{y \in \mathcal{Y}_j} y \, dy / \int_{y \in \mathcal{Y}_j} dy$, as this option can reduce the approximation error (see Appendix for more details).

**Sampling valid causal models.** Monte Carlo (MC) algorithms are a key approach for sampling causal models subject to constraints. However, efficiently sampling probability tables with row and column constraints can be challenging. Denote $\theta_{ijk} = \int_{a \in \mathcal{A}_i, y \in \mathcal{Y}_j, w \in \mathcal{W}_k} dF(a, y, w)$ and $\theta_l = \int_{u \in \mathcal{U}_l} dF(u)$. After discretization, the problem of sampling $x_{ijkl}$ is equivalent to sampling from a simplex $\mathcal{D}$ defined by the linear constraints

$$
\begin{aligned}
&\sum_l x_{ijkl} = \theta_{ijk}, \forall i \in [n_{\mathcal{A}}], j \in [n_{\mathcal{Y}}], k \in [n_{\mathcal{W}}], \\
&\sum_{ijk} x_{ijkl} = \theta_l, \forall l \in [n_{\mathcal{U}}], \\
&0 \le x_{ijkl} \le 1, \forall (i, j, k, l) \in [n_{\mathcal{A}}] \times [n_{\mathcal{Y}}] \times [n_{\mathcal{W}}] \times [n_{\mathcal{U}}].
\end{aligned}
\tag{2}
$$

The overall equalities provide $n_{\mathcal{A}} n_{\mathcal{Y}} n_{\mathcal{W}} + n_{\mathcal{U}} - 1$ linearly independent constraints, as both types of equalities imply $\sum_{ijkl} x_{ijkl} = 1$. Since there are $n_{\mathcal{A}} n_{\mathcal{Y}} n_{\mathcal{W}} n_{\mathcal{U}}$ unknown variables, we need to determine the value of $n_{\mathcal{A}} n_{\mathcal{Y}} n_{\mathcal{W}} n_{\mathcal{U}} - n_{\mathcal{A}} n_{\mathcal{Y}} n_{\mathcal{W}} - n_{\mathcal{U}} + 1$ unknowns.

Directly sampling $x_{ijkl}$ from distributions supported on $[0, 1]$ and rejecting invalid samples that do not satisfy the inequality constraints can be sample-inefficient. To address this issue, we can incorporate inequalities to shrink the support of each unknown variable. For instance, Li and Pearl

(2022) proposed using the bounds $x_{ijkl} \leq \min\{\theta_{ijk}, \theta_l\}$ and $x_{ijkl} \geq \max\{0, \theta_{ijk} + \theta_l - 1\}$ in their optimization problem. However, the support of each unknown $x_{ijkl}$ in (Li and Pearl, 2022) is not as tight as those found by solving the linear programming problem (3):

$$\min / \max x_{ijkl}$$
$$s.t. \quad \mathbf{x} \in \mathcal{D}. \tag{3}$$

Moreover, (Li and Pearl, 2022) only consider constraints of the given action rather than all $a \in \mathcal{A}$, which further contributes to the lack of tightness in their approach. Therefore, the solutions generated by (Li and Pearl, 2022) do not satisfy all the constraints in (2). We provide evidence for this claim by reporting the proportion of valid samples obtained with different sample spaces in Table 5 (see Appendix A.7) for the example discussed in Section 4. We also report the simulation results in Section 4 for causal bounds, which evidently support our arguments.

While the individual bounds for $x_{ijkl}$ obtained by (3) are tight, the sample efficiency is still unsatisfactory. To improve sample efficiency, we introduce the sampling algorithm based on sequential LP. Suppose that we find a linearly independent variable index set $S := \{n_1, n_2, \cdots, n_{|S|}\}$ with cardinality $n_{\mathcal{A}} n_{\mathcal{Y}} n_{\mathcal{W}} n_{\mathcal{U}} - n_{\mathcal{A}} n_{\mathcal{Y}} n_{\mathcal{W}} - n_{\mathcal{U}} + 1$ of (2). For the first variable $x_{n_1}$, we solve (3) to find its support $[l_{n_1}, h_{n_1}]$, and we sample a $\hat{x}_{n_1}$ from a user-given distribution truncated to $[l_{n_1}, h_{n_1}]$. Then, for the $n_i$-step, the previous values of $x_{n_1}, \cdots, x_{n_{i-1}}$ have been determined by sampling. We add the constraints for $x_{n_j}$, $j = 1, 2, \cdots, i-1$, where it is equal to its corresponding value. Together with new constraints in (3), we can find a support $[l_{n_i}, h_{n_i}]$ for $x_{n_i}$ by solving

$$\min / \max x_{n_i}$$
$$s.t. \quad \mathbf{x} \in \mathcal{D}, \tag{4}$$
$$x_{n_j} = \hat{x}_{n_j}, j = 1, 2, \cdots, i-1,$$

where $\hat{x}_{n_j}$ is a sampled value for $x_{n_j}$. After $|S|$ steps, the remaining $x_{ijkl}$ with its index $(i, j, k, l) \notin S$ can be solved by linear equations in (2). These sequential LP steps ensure that we always find valid samples in the simplex defined by (2). Such steps induce a distribution of $\mathbf{x}$ supported on the simplex defined by (3). Although this is the most computationally extensive step, it can significantly improve the sample efficiency. In comparison, the sample efficiency of directly sampling $x_{ijkl}$ from distributions supported on $[0, 1]$ or using the bounds proposed in (Li and Pearl, 2022) is much lower, as shown in Table 5. We will discuss our sampling method on general simplex in Appendix A.

---

**Algorithm 1** Monto-Carlo algorithm for causal bounds with an extra optimization procedure

---

**Input:** cumulative distribution functions $\hat{F}(a, y, w)$ and $\hat{F}(u)$, discrepancy parameter $\epsilon$, sampling distribution $F_s$, batch size $B$, optimization procedure **OPT**

1: Discrete the variable domain $\mathcal{A}, \mathcal{Y}, \mathcal{W}, \mathcal{U}$
2: Select a linearly independent variable index set $S$ with size $n_{\mathcal{A}} n_{\mathcal{Y}} n_{\mathcal{W}} n_{\mathcal{U}} - n_{\mathcal{A}} n_{\mathcal{Y}} n_{\mathcal{W}} - n_{\mathcal{U}} + 1$ for linear equations in (2) and compute each $\hat{\theta}_{ijk}$ and $\hat{\theta}_l$ according to (6)
3: **for** $n = 1, \cdots, B$ **do**
4:     Sample $\theta_x$ from the uniform distribution supported on $[\max\{\hat{\theta}_x - \epsilon, 0\}, \max\{\hat{\theta}_x + \epsilon, 1\}]$ for all $x = ijk$ or $x = l$
5:     Sequentially solve LP (4) to find support $[l_{ijkl}, h_{ijkl}]$ for each $x_{ijkl}$ with $(i, j, k, l) \in S$ and sample a value from $F_s$ supported on $[l_{ijkl}, h_{ijkl}]$
6:     Solving remaining $x_{ijkl}$ by linear equations (2) for all $(i, j, k, l) \notin S$
7:     Find local optima $\mathbf{x}_{loc}^{min}(a)$ and $\mathbf{x}_{loc}^{max}(a)$ via **OPT**(min/max, $\hat{\mathbb{E}}_{\mathcal{M}}[Y|do(a)]$, $\mathcal{D}$, $\mathbf{x}$) for each $a$
8:     Compute the causal effect $b_n(a)$ by the outputs of **OPT** for each $a$
9: For each $a$, sort total $2B$ valid causal bounds and get $b_{(1)}(a), b_{(2)}(a), \cdots, b_{(2B)}(a)$
**Output:** $l(a) = b_{(1)}(a)$ and $h(a) = b_{(2B)}(a)$ for $a \in \mathcal{A}$

---

For any given $a$, the optimization problem (1) after discretization is

$$\max / \min \hat{\mathbb{E}}_{\mathcal{M}}[Y|do(a)]$$
$$s.t. \quad \mathbf{x} \in \mathcal{D}. \tag{5}$$

If all referred random variables are discrete, the optimization (5) can be exactly the same as (1) for natural discretization. To accelerate the convergence speed, we can incorporate an optimization

procedure **OPT**(min/max, object, feasible domain $\mathcal{D}$, initial guess $\mathbf{x}_0$). The optimization procedure **OPT** takes as input the objective function to minimize/maximize, the feasible domain $\mathcal{D}$, and an initial guess $\mathbf{x}_0$. The procedure **OPT** seeks to find local optima $\mathbf{x}_{loc} \in \mathcal{D}$ that optimizes the objective function locally. This assumption on **OPT** is not so strict, because optimization procedures can usually work effectively at local regions and guarantee local optimality for sufficiently close initial guesses. Specifically, in each iteration of Algorithm 1, we use the sampled value $\mathbf{x}$ as the initial guess $\mathbf{x}_0$ and feed it to **OPT**. The output of **OPT** is then used as the updated estimate of causal bounds. Intuitively, **OPT** can make the probability distribution on causal bounds concentrate at local optima rather than spread over the whole feasible domain. By leveraging the optimization procedure, we can prove the result in Proposition 3.1 under mild assumptions. For simplicity, we denote the induced sampling measure for $\mathbf{x}$ as $\mathbb{P}_s$ and the solutions to (5) as $\hat{l}(a)$ and $\hat{h}(a)$.

**Proposition 3.1** *Assume that the sampling measure $\mathbb{P}_s$ satisfies $\forall \mathbf{x} \in \mathcal{D}$, and $\forall \delta > 0$, $\mathbb{P}_s(\mathcal{B}(\mathbf{x}, \delta) \cap \mathcal{D}) > 0$, where $\mathcal{B}(\mathbf{x}, \delta)$ is a ball centered at $\mathbf{x}$ with radius $\delta$. Given a deterministic procedure **OPT** which satisfies for any local optima $\mathbf{x}_{loc}$, there exists $\delta > 0$ such that for any initial guess $\mathbf{x}_0 \in \mathcal{B}(\mathbf{x}_{loc}, \delta) \cap \mathcal{D}$, **OPT** can output $\mathbf{x}_{loc}$ as a result. If the discrepancy parameter $\epsilon$ is set to 0, then $b_{(1)}(a)$ and $b_{(2B)}(a)$ converge almost surely to $\hat{l}(a)$ and $\hat{h}(a)$ for $B \to \infty$, respectively.*

Compared with the concentration property in (Zhang et al., 2022), Proposition 3.1 is more practical in reality. For one thing, the concentration property relies on the distribution of causal bounds, which is often complex and difficult to know in practice. For another, people are usually more interested in convergence results than concentration results in approximation problems. For example, if one has several samples for $l(a)$, it is more reasonable to use the minimum value among them rather than the average results.

**Estimation error.** The remaining concern is the estimation error in RHS of (2). We can constrain or sample $\theta_l$ and $\theta_{ijk}$ according to the known parameters $\hat{\theta}_l$ and $\hat{\theta}_{ijk}$:

$$
\begin{aligned}
\hat{\theta}_{ijk} &= \int_{a \in \mathcal{A}_i, y \in \mathcal{Y}_j, w \in \mathcal{W}_k} d\hat{F}(a, y, w), \forall i \in [n_{\mathcal{A}}], j \in [n_{\mathcal{Y}}], k \in [n_{\mathcal{W}}] \\
\hat{\theta}_l &= \int_{u \in \mathcal{U}_l} d\hat{F}(u), \forall l \in [n_{\mathcal{U}}].
\end{aligned}
\tag{6}
$$

We use the discrepancy parameter $\epsilon$ to reflect the estimation in Algorithm 5. For detailed discussion, we refer readers to Appendix A.

**MAB.** We now consider the first transfer learning task for $|\mathcal{A}| < \infty$. The algorithm proceeds as follows: first, we remove any arm $a$ for which $h(a) < \max_{i \in \mathcal{A}} l(i)$. Next, we truncate the upper confidence bound $U_a(t)$ for the remaining actions using their respective causal bounds. Specifically, we define the truncated upper confidence bound as $\hat{U}_a(t) = \min\{U_a(t), h(a)\}$. The algorithm then chooses the action with the highest truncated UCB and update the average reward of the chosen action. Since the MAB can be regarded as the contextual bandit with $|\mathcal{W}| = 1$, our Algorithm 6 in appendix is a special of Algorithm 2. As a simple corollary of Theorem A.3, the minimax upper bound is $\mathcal{O}(\sqrt{|\widetilde{\mathcal{A}^*}|T \log T})$, which also matches the lower bound $\Omega(\sqrt{|\widetilde{\mathcal{A}^*}|T})$ in transfer learning.

### 3.2 Transfer learning in partially observable contextual bandit

We will now consider a more challenging scenario in which knowledge is transferred between two partially observable contextual bandit agents. In this task, the agent need to minimize the contextual regrets given partial model knowledge $\hat{F}(a, y, w)$ and $\hat{F}(u)$. Similarly, we need to solve the following optimization problem:

$$
\begin{aligned}
&\sup / \inf \mathbb{E}_{\mathcal{M}}[Y | do(a), w] \\
s.t. \quad &F_{\mathcal{M}}(a, y, w) = F(a, y, w), \\
&F_{\mathcal{M}}(u) = F(u),
\end{aligned}
\tag{7}
$$

and discrete it as we do in Section 3.1. The object is $\hat{\mathbb{E}}_{\mathcal{M}}[Y | do(a_i), w_k] = \sum_{j,l} \frac{y_j \theta_l x_{ijkl}}{\sum_{j'} x_{ij'kl}}$. Algorithm 7 provides causal bounds for several discrete points, which can be generalized to the entire space $\mathcal{Y} \times \mathcal{W}$ through interpolation.

**Contextual bandit.** We will begin by discussing discrete contexts and then move on to more general function approximation problems. The objective is to minimize the contextual bandit regret. Suppose that the true causal effect $\mathbb{E}[Y|do(a), w]$ falls within the interval $[l(w, a), h(w, a)]$. We define the set $\mathcal{A}^*(x)$ as follows: $\mathcal{A}^*(x) = \mathcal{A} - \{a \in \mathcal{A}|h(x, a) \leq \max_{i \in \mathcal{A}} l(x, i)\}$. This set eliminates suboptimal context-action pairs $(w, a)$ through causal bounds.

---

**Algorithm 2** Transfer learning in contextual bandit

---

**Input:** time horizon $T$, causal bound $l(w, a)$ and $h(w, a)$
 1: Initialize the empirical mean $\mu_{w,a}(1)$ and the number of pulling $n_{w,a}(1)$ to zero with size $|\mathcal{A}||\mathcal{W}|$
 2: **for** round $t = 1, 2, \cdots, T$ **do**
 3:    Observe the context $w_t$
 4:    Compute the upper confidence bound $U_{w_t,a}(t) = \min\{1, \hat{\mu}_{w_t,a}(t) + \sqrt{\frac{\ln t}{n_{w_t,a}(t)}}\}$
 5:    Compute the best arm candidate set $\mathcal{A}^*(w_t)$
 6:    Truncate $U_{w_t,a}(t)$ to $\hat{U}_{w_t,a}(t) = \min\{U_{w_t,a}(t), h(w_t, a)\}$ for $a \in \mathcal{A}^*(w_t)$
 7:    Take the action $a_t = \arg\max_{a \in \mathcal{A}^*(w_t)} \hat{U}_{w_t,a}(t)$ and observe a reward $y_t$
 8:    Update the empirical mean $\hat{\mu}_{w_t,a}(t)$ and the number of pulling $n_{w_t,a}(t)$

---

**Theorem 3.1** *Consider a contextual bandit problem with $|\mathcal{A}| < \infty$ and $|\mathcal{W}| < \infty$. For each arm $a \in \mathcal{A}$ and expected conditional reward $\mu_{w,a}$ bounded by $[l(w, a), h(w, a)]$. For any given context $w \in \mathcal{W}$, suppose $w$ occurs for $T_w$ times. Then in the Algorithm 2, the conditional number of draws $\mathbb{E}[N_a(T_w)]$ for any sub-optimal arm is upper bounded as: (1) 0, if $h(w, a) < \max_{i \in \mathcal{A}} l(w, i)$; (2) $\frac{|\mathcal{A}|\pi^2}{6}$, if $\max_{i \in \mathcal{A}} l(w, i) \leq h(w, a) < \mu_w^*$; (3) $\frac{8 \log T_w}{\Delta_{w,a}^2} + |\mathcal{A}|$, if $h(w, a) \geq \mu_w^*$.*

Theorem 3.1 shows how causal bounds improve the performance of classical bandit algorithms by controlling the number of times each arm is pulled. This theorem also generalizes that in (Zhang and Bareinboim, 2017) as well as Theorem A.2, because MAB is a special case when $|W| = 1$.

The regret of Algorithm 2 scale with $\mathcal{O}(\sum_{w \in \mathcal{W}} \sqrt{|\widetilde{\mathcal{A}^*}(w)|\mathbb{P}(W = w)T \log T})$ (see Table 1 and Theorem A.3), where $\widetilde{\mathcal{A}^*}(x) := \mathcal{A} - \{a \in \mathcal{A}|h(x, a) < \mu_w^*\}$. Although our Algorithm 2 does not explicitly remove suboptimal arms in $\widetilde{\mathcal{A}^*}(w) - \mathcal{A}^*(w)$, the property of UCB can still control the number of such suboptimal arms due to truncation by causal upper bounds. Using Cauchy-Schwarz inequality, we have $\sum_{w \in \mathcal{W}} \sqrt{8(|\widetilde{\mathcal{A}^*}(w)| - 1)\mathbb{P}(W = w)} \leq \sqrt{8 \sum_{w \in \mathcal{W}}(|\widetilde{\mathcal{A}^*}(w)| - 1)}$. This inequality shows clearly that any elimination of suboptimal arms will improve transfer learning algorithms. It also implies that our regret result outperform that of the generic RL methods in terms of minimax versions, as the minimax regret in (Dann et al., 2021) scales with the right hand side.

Denote the contextual bandit instances with prior knowledge $l(w, a)$ and $h(w, a)$ as $\mathfrak{M} = \{$contextual bandit instances with $l(w, a) \leq \mu_{w,a} \leq h(w, a), \forall a \in \mathcal{A}, \forall w \in \mathcal{W}\}$. The following lower bound result indicates that our Algorithm 2 is near-optimal up to logarithmic terms in $T$.

**Theorem 3.2** *Suppose $|\mathcal{A}| < \infty$ and $|\mathcal{W}| < \infty$. Then for any algorithm* A*, there exists an instance $\nu \in \mathfrak{M}$ such that $\liminf_{T \to \infty} \frac{Reg(T)}{\sqrt{T}} \geq \frac{1}{27} \sum_{w \in \mathcal{W}} \sqrt{(|\widetilde{\mathcal{A}^*}(w)| - 1)\mathbb{P}(W = w)}$.*

**Function approximation.** Next, we consider transfer learning of contextual bandits in function approximation setting. Once again, we assume that the true causal effect falls within $[l(w, a), h(w, a)]$. We eliminates the functions that cannot be the true reward function using causal bounds: $\mathcal{F}^* = \{f \in \mathcal{F}|l(w, a) \leq f(w, a) \leq h(w, a)\}$ and modify the best arm candidate set as following: $\mathcal{A}^*(x) = \{a \in \mathcal{A}|a = \arg\max_{i \in \mathcal{A}} f(x, i)$ for some $f$ in $\mathcal{F}^*\}$. Our Algorithm 3 is based on the inverse gap weighting technique (IGW) (Foster et al., 2018; Agarwal et al., 2012; Foster et al., 2020; Simchi-Levi and Xu, 2021). In previous literature concerning IGW, this strategy is employed using a fixed action set size $|\mathcal{A}|$, and the learning rate $\gamma$ is typically chosen to follow fixed non-adaptive schedule. However, our algorithm differs from these existing approaches in three aspects. Firstly, we only apply the IGW scheme to functions and actions that are not eliminated by causal bounds. Secondly, we select the learning rate $\gamma_t$ to be context-dependent, capturing the influence of causal bounds on the arm set. This adaptive approach enhances the efficiency of our algorithm in the online learning process,

resulting in improved performance in the transfer learning task. Thirdly, in comparison with (Foster et al., 2020), our regret order in $T$ is $\mathcal{O}(\sqrt{T \log(\delta^{-1} \log T)})$ instead of $\mathcal{O}(\sqrt{T \log(\delta^{-1} T^2)} \log T)$. By eliminating suboptimal functions using certain causal bounds rather than relying on data-driven upper confidence bounds on the policy space, we are able to save additional $\log T$ terms.

---

**Algorithm 3** Transfer learning in contextual bandits with function approximation

---

**Input:** time horizon $T$, tuning parameters $\delta, \eta$, causal bounds $l(w, a)$ and $h(w, a)$
1: Eliminate function space $\mathcal{F}$ and obtain $\mathcal{F}^*$ via causal bounds
2: Set epoch schedule $\{\tau_m = 2^m, \forall m \in \mathbb{N}\}$
3: **for** epoch $m = 1, 2, \cdots, \lceil \log_2 T \rceil$ **do**
4:      Find the function $\hat{f}_m = \arg\min_{f \in \mathcal{F}^*} \sum_{t=1}^{\tau_{m-1}} (f(w_t, a_t) - y_t)^2$ via the **Least Square Oracle**
5:      **for** round $t = \tau_{m-1} + 1, \cdots, \tau_m$ **do**
6:          Observe the context $w_t$
7:          Compute the best action candidate set $\mathcal{A}^*(w_t)$
8:          Compute $\gamma_t = \sqrt{\frac{\eta |\mathcal{A}^*(w_t)| \tau_{m-1}}{\log(2\delta^{-1}|\mathcal{F}^*| \log T)}}$ (for the first epoch, $\gamma_1 = 1$)
9:          Compute $\hat{f}_m(w_t, a)$ for each action $a \in \mathcal{A}^*(w_t)$ and the following probabilities

$$p_t(a) = \begin{cases} \frac{1}{|\mathcal{A}^*(w_t)| + \gamma_t(\hat{f}_m(w_t, \hat{a}_t) - \hat{f}_m(w_t, a))}, & \text{for all } a \in \mathcal{A}^*(w_t) - \{\hat{a}_t\} \\ 0, & \text{for all } a \in \mathcal{A} - \mathcal{A}^*(w_t), \\ 1 - \sum_{a \neq \hat{a}_t} p_t(a), & \text{for } a = \hat{a}_t, \end{cases}$$

     where $\hat{a}_t = \max_{a \in \mathcal{A}} \hat{f}_m(w_t, a)$.
10:          Sample $a_t \sim p_t(\cdot)$ and observe a reward $y_t$

---

**Theorem 3.3** *Consider a contextual bandit problem with $|\mathcal{A}| < \infty$ and $|\mathcal{F}| < \infty$. Suppose that the realizability Assumption 2.1 holds. Then with probability at least $1 - \delta$, the expected regret $\mathbb{E}[Reg(T)]$ of Algorithm 3 is upper bounded by $\mathcal{O}\left(\sqrt{\mathbb{E}_W[\mathcal{A}^*(W)] T \log(\delta^{-1}|\mathcal{F}^*| \log T)}\right)$.*

Previous works such as (Zhang and Bareinboim, 2021; Liu et al., 2021) have also explored transfer learning in general contextual bandits, but their regrets scale with $\sqrt{|\Pi|}$ instead of the more desirable $\sqrt{\log |\Pi|}$. Specifically, they treat each basis policy as an independent arm in bandit problems, which is often invalid as similar policies can provide information to each other. For example, exploring the optimal policy can also provide information about the second optimal policy, since these two policies typically yield similar actions and only differ in a small amount of contexts. This insight helps explain why their regret depends on $\sqrt{|\Pi|}$ (as $|\Pi| = |\mathcal{F}|$) rather than more desirable $\sqrt{\log |\Pi|}$. Additionally, the method in Zhang and Bareinboim (2021) relies on instrumental variables, but our method does not rely on them.

We also prove that our upper bound in Theorem 3.3 matches the lower bound in transfer learning. Denote the contextual bandit instances with prior knowledge $l(w, a)$ and $h(w, a)$ as $\mathfrak{M} = \{$contextual bandit instances with $l(w, a) \leq f^*(w, a) \leq h(w, a), \forall (a, w) \in \mathcal{A} \times \mathcal{W}\}$.

**Theorem 3.4** *Consider a contextual bandit problem with $|\mathcal{A}| < \infty$ and $|\mathcal{F}| < \infty$. Suppose that the agent has access to the function space $\mathcal{F}$ and the realizability assumption holds. Then for any algorithm $\mathsf{A}$, there exists an instance $\nu \in \mathfrak{M}$ such that $\liminf_{T \to \infty} \frac{Reg(T)}{\sqrt{T}} \geq \sqrt{\mathbb{E}_W[|\mathcal{A}^*(W)|] \log |\mathcal{F}^*|}$.*

To conclude this section, we summarize our theoretical results in Table 1.

## 4 Numerical Experiments

In this section, we conduct experiments to validate our findings. For detailed experimental setup, we refer readers to Appendix A.7.

**Causal bounds.** To evaluate the effectiveness of our proposed Algorithm 5, we compare it with the method proposed by Li and Pearl (2022) when all variables are binary. The results are presented

| Transfer Learning Tasks | Gap-dependent Upper Bound | Minimax Upper Bound |
|---|---|---|
| multi-armed bandit | $\sum_{a \in \mathcal{A}, \Delta_a > 0} \frac{\log T}{\Delta_a}$ | $\sqrt{|\mathcal{A}| T \log T}$ |
| | $\sum_{a \in \widetilde{\mathcal{A}^*}, \Delta_a > 0} \frac{\log T}{\Delta_a}$ | $\sqrt{|\widetilde{\mathcal{A}^*}| T \log T}$ |
| contextual bandits with finite contexts | $\sum_{w \in \mathcal{W}} \sum_{a \in \mathcal{A}, \Delta_{w,a} > 0} \frac{\log T}{\Delta_{w,a}}$ | $\sqrt{|\mathcal{W}||\mathcal{A}| T \log T}$ |
| | $\sum_{w \in \mathcal{W}} \sum_{a \in \widetilde{\mathcal{A}^*}(w), \Delta_{w,a} > 0} \frac{\log T}{\Delta_{w,a}}$ | $\sum_{w \in \mathcal{W}} \sqrt{|\widetilde{\mathcal{A}^*}(w)| \mathbb{P}(W = w) T \log T}$ |
| contextual bandits with function approximation | not applicable | $\sqrt{|\mathcal{A}| T \log(\delta^{-1}|\mathcal{F}| \log T)}$ |
| | | $\sqrt{\mathbb{E}_W[|\mathcal{A}^*(W)|] T \log(\delta^{-1}|\mathcal{F}^*| \log T)}$ |

| Transfer Learning Tasks | Minimax Lower Bound | Note |
|---|---|---|
| multi-armed bandit | $\Omega\left(\sqrt{|\mathcal{A}| T}\right)$ | $\widetilde{\mathcal{A}^*} := \{a \in \mathcal{A} | h(a) \geq \mu^*\} \subset \mathcal{A}$ |
| | $\Omega\left(\sqrt{|\widetilde{\mathcal{A}^*}| T}\right)$ | |
| contextual bandits with finite contexts | $\Omega\left(\sqrt{|\mathcal{W}||\mathcal{A}| T}\right)$ | $\sum_{w \in \mathcal{W}} \sqrt{|\widetilde{\mathcal{A}^*}(w)| \mathbb{P}(W = w)}$ |
| | $\Omega\left(\sum_{w \in \mathcal{W}} \sqrt{|\widetilde{\mathcal{A}^*}(w)| \mathbb{P}(W = w) T}\right)$ | $\leq \sqrt{\sum_{w \in \mathcal{W}} |\mathcal{A}^*(w)|}$ |
| contextual bandits with function approximation | $\Omega\left(\sqrt{|\mathcal{A}| T \log(|\mathcal{F}|)}\right)$ | $\mathcal{F}^* \subset \mathcal{F}$ and |
| | $\Omega\left(\sqrt{\mathbb{E}_W[|\mathcal{A}^*(W)|] T \log(|\mathcal{F}^*|)}\right)$ | $\mathcal{A}^*(w) \subset \mathcal{A}$ for each $w \in \mathcal{W}$ |

Table 1: Comparison with classical bandit algorithms (row 2,4,6) and ours (row 3,5,7). Here, we hide the absolute constants. As for notations, readers can refer to Appendix A.1 for a quick review.

in Table 2. Our method can tighten the bounds obtained by Li and Pearl (2022), as evidenced by the narrower ranges of the estimated causal effects. Moreover, we observe that solving non-linear optimization is quite unstable and often leads to local optima. To obtain global solutions, we repeat solving non-linear optimization with randomly initialized starting points.

| causal effect | non-linear optimization (Li and Pearl, 2022) | our Algorithm 5 |
|---|---|---|
| $\mathbb{E}[Y|do(A = 0)]$ | $[0.283, 0.505]$ | $[0.371, 0.466]$ |
| $\mathbb{E}[Y|do(A = 1)]$ | $[0.240, 0.807]$ | $[0.300, 0.705]$ |

Table 2: Comparison with causal bounds in partially observable problems.

**Transfer learning in MAB.** We compare our proposed Algorithm 6 with standard multi-armed bandit (MAB) algorithms that do not have access to causal bounds and the method (CUCB) in (Zhang and Bareinboim, 2017). We also include the counterparts that incorporate a naive transfer procedure (without distinguishing the do-distribution), which we refer to as UCB-.

The regret curves are presented in Figure 2(Left). We observe that the regret of UCB- sharply grows linearly and becomes sublinear after a certain turning point. This negative transfer result indicates that the agent spends many trials to overcome the influence of wrong knowledge. In contrast, our proposed algorithm achieves orders of magnitude faster convergence rates, as the causal bounds effectively eliminate suboptimal arms. These results corroborate with our findings and demonstrate that prior experiences can be transferred to improve the performance of the target agent. Moreover, we find that a tighter causal bound can further improve the performance of classical algorithms. Overall, our experiments provide strong evidence that our proposed transfer learning algorithm can significantly improve the performance of MAB agents.

**Transfer learning in contextual bandits.** We compare the performance of our algorithm with FALCON (Simchi-Levi and Xu, 2021), a well-known version of IGW. From the numerical results in Figure 2(Right), we observe that our algorithm outperforms FALCON significantly, even without eliminating infeasible functions. The average size of $\mathcal{A}^*(w)$ is 3.254 in our randomly selected instances, indicating that eliminating action size can significantly improve performance. Additionally, our algorithm is particularly effective for homogeneous functions, as these functions usually attain the maximum at the same points. In this case, adaptively eliminating suboptimal actions is a very effective way to reduce regrets.

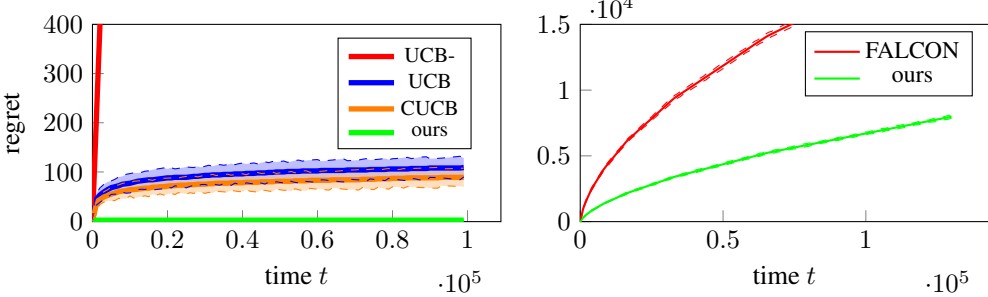

Fig. 2: Comparison with classical and causally enhanced algorithms. The top and bottom dashed curves represent the curves obtained by adding and subtracting one standard deviation to the regret curve of the corresponding color, respectively. Left: transfer learning in MAB; Right: transfer learning in contextual bandit.

# 5 ADDITIONAL ONE PAGE

## 5.1 CONCLUSION

In this paper, we investigate transfer learning in partially observable contextual bandits by converting the problem to identifying or partially identifying causal effects between actions and rewards. We derive causal bounds with the existence of partially observable confounders using our proposed Monte-Carlo algorithms. We formally prove and empirically demonstrate that our causally enhanced algorithms outperform classical bandit algorithms and achieve orders of magnitude faster convergence rates.

There are several future research directions we wish to explore. Firstly, we aim to investigate whether the solutions to discretization optimization converge to those of the original problem. While the approximation error can be exactly zero when all referred random variables are discrete, it is still unclear which conditions for general random variables and discretization methods can lead to convergence. We conjecture that this property may be related to the sensitivity of the non-linear optimization problem.

Lastly, we aim to extend our IGW-based algorithm to continuous action settings. IGW has been successfully applied to continuous action settings and has shown practical advantages in large action spaces. This extension may be related to complexity measures in machine learning.

## 5.2 STRUCTURE OF APPENDIX

In appendix A, we put the omitted information regarding the examples, algorithms, theorems, implementation details and numerical setup. We also put partial related work section due to the strict page limitation of ICLR. In appendix B, you can find all proofs about claimed theorems. In appendix C, you can find some related materials about causal tools. In appendix D, we generalize our sampling method on general simplex more than the special one defined by the special optimization (1). In appendix E, we put a conclusion section and discuss the future work.

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

## A OMITTED RESULTS

### A.1 NOTATIONS

To help readers understand our paper better. We restate the important notations in our table 1.

1. $\mathcal{A}$: the action space
2. $\mathcal{W}$: the context space where the context can be observed by the learner
3. $\mathcal{W}$: the context space where the context cannot be observed by the learner
4. $T$: time horizon
5. $h(a)$: causal upper bound for $\mathbb{E}[Y|do(a)]$
6. $l(a)$: causal lower bound for $\mathbb{E}[Y|do(a)]$
7. $\mathcal{F}$: function space
8. $\Pi$: policy space induced by the given function space
9. $\mathcal{F}^*$: the truncated function space
10. $\delta$: confidence parameter
11. $\mu^*$: the maximal mean among actions
12. $\Delta_a$: the gap for action $a$
13. $\Delta_{w,a}$: the gap for action $a$ when the context $w$ occurs
14. $\widetilde{\mathcal{A}^*}$: the action set truncated by the causal bounds in MAB setting
15. $\widetilde{\mathcal{A}^*}(w)$: the action set truncated by the causal bounds in tabular contextual bandit setting
16. $\mathcal{A}^*(w)$: the action set truncated by the causal bounds in contextual bandit setting with function approximation
17. $\mathbb{P}(W = w)$: PMF of context $W$

### A.2 TABLES OF THE EXAMPLE IN SECTION 2

| $(U, W)$ | $(0,0)$ | $(0,1)$ | $(1,0)$ | $(1,1)$ |
|---|---|---|---|---|
| $\mathbb{E}[Y|do(A=0)]$ | 0 | 0 | 1 | 1 |
| $\mathbb{E}[Y|do(A=1)]$ | 10 | 10 | 0.9 | 0.9 |

Table 3: Reward table

| $(A, W)$ | $(0,0)$ | $(0,1)$ | $(1,0)$ | $(1,1)$ |
|---|---|---|---|---|
| $\mathbb{P}(A,W)$ | 0.05 | 0.05 | 0.45 | 0.45 |

Table 4: Observation probability for agents

### A.3 INCORPORATING ESTIMATION ERROR

Since $\theta_{ijk} \in [0,1]$, the variance of one sample is at most $\frac{1}{4}$. If $\theta_{ijk}$ has $n$ i.i.d. samples, then its largest variance of the known $\hat{\theta}_{ijk}$ is $\frac{1}{4n}$. We can simply set $\epsilon = \frac{1}{2\sqrt{n}}$ if $n$ is given. For example, if one want to reflect the concentration property, then one can choose the truncate Gaussian distribution $\mathcal{N}(\hat{\theta}_x, \frac{1}{4n_x})$ for $x = ijk$ or $x = l$. If one expects the worst cases, then one can choose the uniform distribution for $\theta_x$ with discrepancy parameter $\epsilon$.

### A.4 REFERRED ALGORITHMS

We generalize the sampling method for general simplex $\mathcal{D}$

$$A\mathbf{x} \leq \mathbf{b}, \mathbf{x} \geq \mathbf{0}.$$

in Algorithm 4. Without loss of generality, we always assume $\mathcal{D}$ is not empty.

---

**Algorithm 4** A sampling algorithm for the given simplex

---

**Input:** a simplex $\mathcal{D}$, a sampling distribution $F_s$ supported on $[0, 1]$
 1: Denote the number of components of $\mathbf{x}$ as $d$
 2: Solving the following LP

$$\min / \max x_1$$
$$A\mathbf{x} \leq \mathbf{b}, \mathbf{x} \geq \mathbf{0}.$$

  to find the bound $[l_1, h_1]$ for $x_1$
 3: Sample a value $\hat{x}_1$ from the truncated $F_s$ supported on $[l_1, h_1]$
 4: **for** $i = 2, \cdots, d$ **do**
 5:   Solving the follow LP

$$\min / \max x_i$$
$$A\mathbf{x} \leq \mathbf{b}, \mathbf{x} \geq \mathbf{0}$$
$$x_j = \hat{x}_j, j = 1, \cdots, i - 1.$$

  to find the bound $[l_i, h_i]$ for $x_1$
 6:   Sample a value $\hat{x}_i$ from the truncated $F_s$ supported on $[l_i, h_i]$
**Output:** a valid sample $\hat{\mathbf{x}} = (\hat{x}_1, \cdots, \hat{x}_d) \in \mathcal{D}$

---

**Algorithm 5** MC for causal bound

---

**Input:** cumulative distribution functions $\hat{F}(a, y, w)$ and $\hat{F}(u)$, discrepancy parameter $\epsilon$, sampling distribution $F_s$, batch size $B$
 1: Discrete the variable domain $\mathcal{A}, \mathcal{Y}, \mathcal{W}, \mathcal{U}$
 2: Select a linearly independent variable index set $S$ with size $n_{\mathcal{A}} n_{\mathcal{Y}} n_{\mathcal{W}} n_{\mathcal{U}} - n_{\mathcal{A}} n_{\mathcal{Y}} n_{\mathcal{W}} - n_{\mathcal{U}} + 1$ for linear equations (2)
 3: Compute each $\hat{\theta}_{ijk}$ and $\hat{\theta}_l$ according to (6)
 4: **for** $n = 1, 2 \cdots, B$ **do**
 5:   Sample $\theta_x$ from the uniform distribution supported on $[\max\{\hat{\theta}_x - \epsilon, 0\}, \max\{\hat{\theta}_x + \epsilon, 1\}]$ for all $x = ijk$ or $x = l$
 6:   Sequentially solve LP (4) to find support $[l_{ijkl}, h_{ijkl}]$ for each $x_{ijkl}$ with $(i, j, k, l) \in S$ and sample a value from $F_s$ truncated to $[l_{ijkl}, h_{ijkl}]$
 7:   Solving remaining $x_{ijkl}$ by linear equations (2) for all $(i, j, k, l) \notin S$
 8:   Compute the causal effect $b_n(a)$ by $\hat{\mathbb{E}}_{\mathcal{M}}[Y|do(a)]$ for each $a$
 9: For each $a$, sort $B$ valid causal bound and get $b_{(1)}(a), b_{(2)}(a), \cdots, b_{(B)}(a)$
**Output:** $l(a) = b_{(1)}(a)$ and $h(a) = b_{(B)}(a)$ for $a \in \mathcal{A}$

---

## A.5 REFERRED THEOREMS

**Theorem A.1** *Given a causal diagram $\mathcal{G}$ and a distribution compatible with $\mathcal{G}$, let $\{W, U\}$ be a set of variables satisfying the back-door criterion in $\mathcal{G}$ relative to an ordered pair $(A, Y)$, where $\{W, U\}$ is partially observable, i.e., only probabilities $\hat{F}(a, y, w)$ and $\hat{F}(u)$ with the maximum estimation error $\epsilon$, the causal effects of $A$ on $Y$ are then bounded as follows:*

$$l(a_0) \leq \mathbb{E}[Y|do(a_0)] \leq h(a_0),$$

---

**Algorithm 6** Transfer learning in multi-armed bandit

---

**Input:** time horizon $T$, causal bound $l(a)$ and $h(a)$

1: Remove the arm $a$ for $h(a) < \max_{i \in \mathcal{A}} l(i)$ and denote the remaining arm set as $\mathcal{A}^*$
2: Initialize reward vector $\hat{\mu}_a(1)$ and the number of pull $n_a(1)$ to zero, for all $a \in \mathcal{A}^*$
3: **for** round $t = 1, 2, \cdots, T$ **do**
4:  Compute the upper confidence bound $U_a(t) = \min\{1, \hat{\mu}_a(t) + \sqrt{\frac{2 \log T}{n_a(t)}}\}$
5:  Truncate $U_a(t)$ to $\hat{U}_a(t) = \min\{U_a(t), h(a)\}$ for all $a \in \mathcal{A}^*$
6:  Choose the action $a_t = \arg\max_{a \in \mathcal{A}^*} \hat{U}_a(t)$ and observe a reward $y_t$
7:  Update the empirical mean $\hat{\mu}_{a_t}(t+1) = \frac{\hat{\mu}_{a_t} n_{a_t}(t) + y_t}{n_{a_t}(t)+1}$ and the number of pulling $n_{a_t}(t+1) = n_{a_t}(t) + 1$
8:  For $a \neq a_t$, update $\hat{\mu}_a(t+1) = \hat{\mu}_a(t)$ and $n_a(t+1) = n_a(t)$

---

**Algorithm 7** MC for causal bound with $w$

---

**Input:** cumulative distribution functions $\hat{F}(a, y, w)$ and $\hat{F}(u)$, discrepancy parameter $\epsilon$, sampling distribution $F_s$, batch size $B$

1: Discrete the variable domain $\mathcal{A}, \mathcal{Y}, \mathcal{W}, \mathcal{U}$
2: Select a linearly independent variable index set $S$ with size $n_\mathcal{A} n_\mathcal{Y} n_\mathcal{W} n_\mathcal{U} - n_\mathcal{A} n_\mathcal{Y} n_\mathcal{W} - n_\mathcal{U}$ for linear equations (2)
3: Compute each $\hat{\theta}_{ijk}$ and $\hat{\theta}_l$ according to (6)
4: **for** $n = 1, 2, \cdots, B$ **do**
5:  Sample $\theta_x$ from the uniform distribution supported on $[\max\{\hat{\theta}_x - \epsilon, 0\}, \max\{\hat{\theta}_x + \epsilon, 1\}]$ for all $x = ijk$ or $x = l$
6:  Sequentially solve LP (4) to find support $[l_{ijkl}, h_{ijkl}]$ for each $x_{ijkl}$ with $(i, j, k, l) \in S$ and sample a value from $F_s$ supported on $[l_{ijkl}, h_{ijkl}]$
7:  Solving remaining $x_{ijkl}$ by linear equations (2) for all $(i, j, k, l) \notin S$
8:  Compute the causal effect $b_n(w, a)$ by $\hat{\mathbb{E}}_\mathcal{M}[Y|do(a), w]$ for each $a$ and $w$
9: For each $w, a$, sort $B$ valid causal bound and get $b_{(1)}(w, a), b_{(2)}(w, a), \cdots, b_{(B)}(w, a)$

**Output:** $l(w, a) = b_{(1)}(w, a)$ and $h(w, a) = b_{(B)}(w, a)$ for $(w, a) \in \mathcal{W} \times \mathcal{A}$

---

*where $l(a_0)$ and $h(a_0)$ are solutions to the following functional optimization problem for any given $a_0$*

$$l(a_0) = \inf \int_{w \in \mathcal{W}, u \in \mathcal{U}} \int_{y \in \mathcal{Y}} y \, dF(y|a_0, w, u) dF(w, u)$$

$$h(a_0) = \sup \int_{w \in \mathcal{W}, u \in \mathcal{U}} \int_{y \in \mathcal{Y}} y \, dF(y|a_0, w, u) dF(w, u)$$

$$s.t. \int_{u \in \mathcal{U}} dF(a, y, w, u) = F(a, y, w), \forall (a, y, w) \in \mathcal{A} \times \mathcal{W} \times \mathcal{U}$$

$$\int_{a \in \mathcal{A}, y \in \mathcal{Y}, w \in \mathcal{W}} dF(a, y, w, u) = F(u), \forall u \in \mathcal{U}$$

$$\int_{y \in \mathcal{Y}} dF(a, y, w, u) = F(a, w, u), \forall (a, w, u) \in \mathcal{A} \times \mathcal{W} \times \mathcal{U}$$

$$\int_{a \in \mathcal{A}} dF(a, w, u) du = F(w, u), \forall (w, u) \in \mathcal{W} \times \mathcal{U}$$

$$F(y|a, w, u) F(a, w, u) = F(a, y, w, u), \forall (a, y, w, u) \in \mathcal{A} \times \mathcal{Y} \times \mathcal{W} \times \mathcal{U}$$

$$|F(a, y, w) - \hat{F}(a, y, w)| \leq \epsilon, \forall (a, y, w) \in \mathcal{A} \times \mathcal{Y} \times \mathcal{W}$$

$$|F(u) - \hat{F}(u)| \leq \epsilon, \forall u \in \mathcal{U}.$$

*Here the inf/sup is taken with respect to all unknown cumulative distribution functions $F(a, y, w, u)$, $F(a, w, u)$, $F(y|a, w, u)$, $F(w, u)$, $F(a, y, w)$ and $F(u)$.*

**Theorem A.2** *Consider a $|\mathcal{A}|$-MAB problem with rewards bounded in $[0, 1]$. For each arm $a \in \mathcal{A}$, its expected reward $\mu_a$ is bounded by $[l(a), h(a)]$. Then in the Algorithm 6, the number of draws $\mathbb{E}[N_a(T)]$ for any sub-optimal arm is upper bounded as:*

$$
\mathbb{E}[N_a(T)] \leq \begin{cases} 0, h(a) < \max_{i \in \mathcal{A}} l(i) \\ \dfrac{\pi^2}{6}, \max_{i \in \mathcal{A}} l(i) \leq h(a) < \mu^* \\ \dfrac{\log T}{\Delta_a^2}, h(a) \geq \mu^*. \end{cases}
$$

**Theorem A.3** *Consider a contextual bandit problem with $|\mathcal{A}| < \infty$ and $|\mathcal{W}| < \infty$. Denote*

$$
\widetilde{\mathcal{A}^*}(x) = \mathcal{A} - \{a \in \mathcal{A} | h(x, a) < \mu_w^*\}.
$$

*Then the regret of Algorithm 2 satisfies*

$$
\limsup_{T \to \infty} \frac{\mathbb{E}[Reg(T)]}{\sqrt{T \log T}} \leq \sum_{w \in \mathcal{W}} \sqrt{8(|\widetilde{\mathcal{A}^*}(w)| - 1)\mathbb{P}(W = w)}.
$$

## A.6 Implementation details of Algorithm 3

In Algorithm 3, one needs to compute $\mathcal{F}^*$ and $\mathcal{A}^*(w)$. A naive way costs $\mathcal{O}(|\mathcal{F}|)$ time complexity, which becomes inefficient for large $|\mathcal{F}|$ and infeasible for infinite $|\mathcal{F}|$. Actually, we can implicitly compute $\mathcal{F}^*$ by clipping, i.e., using $\min\{\max\{\hat{f}_m(w, a), l(w, a)\}, h(w, a)\}$ as the estimator at the epoch $m$. As $\hat{f}_m$ gets closer to the true reward function $f^*$, which is within the causal bounds, the causal bounds gradually lose their constraint effect. For computing $\mathcal{A}^*(w)$, we refer readers to the section 4 of (Foster et al., 2020), where a systematic method for computing $\mathcal{A}^*(w)$ within a given accuracy is provided.

Another option to implement Algorithm 3 is to compute $\mathbb{E}_W[|\mathcal{A}^*(W)|]$ using expert knowledge $F(a, y, w)$. We can set $\gamma_t = \sqrt{\frac{\eta \mathbb{E}_W[|\mathcal{A}^*(W)|]\tau_{m-1}}{\log(2\delta^{-1}|\mathcal{F}^*| \log T)}}$, so that $\gamma_t$ remains constant within an epoch. Our proof still holds for this option, and the regret order is the same as in Theorem 3.3. Intuitively, $|\mathcal{A}(w_t)|$ is a sample from an induced distribution with a mean of $\mathbb{E}_W[|\mathcal{A}^*(W)|]$, so on average, the regrets of both options are of the same order.

It is worth noting that Algorithm 3 and Theorem 3.3 can be easily extended to handle infinite $\mathcal{F}$ using standard learning-theoretic tools such as metric entropy. Suppose $\mathcal{F}$ is equipped with a maximum norm $\|\cdot\|$. We can consider an $\epsilon$-covering $\mathcal{F}_\epsilon^*$ of $\mathcal{F}^*$ under maximum norm. Since $|\mathcal{F}_\epsilon^*|$ is finite, we can directly replace $\mathcal{F}^*$ with $\mathcal{F}_\epsilon^*$ and do not change any algorithmic procedure. Thanks to the property of $\epsilon$-covering, there exists a function $f_\epsilon^* \in \mathcal{F}_\epsilon^*$ such that $\|f_\epsilon^* - f^*\| \leq \epsilon$. Hence, the regret can be bounded by

$$
Reg(T) \leq 8\sqrt{\mathbb{E}_W[\mathcal{A}^*(W)]T \log(2\delta^{-1}|\mathcal{F}_\epsilon^*| \log T)} + \epsilon T.
$$

By replacing the dependence on $\log |\mathcal{F}^*|$ in the algorithm's parameters with $\log |\mathcal{F}_\epsilon^*|$ and setting $\epsilon = \frac{1}{T}$, we obtain a similar result as Theorem 3.3 up to an additive constant of 1.

**Definition A.1 ((Fan, 1953))** *Let $(\mathcal{F}, \|\cdot\|)$ be a normed space. The set $\{f_1, \cdots, f_N\}$ is an $\epsilon$-covering of $\mathcal{F}$ if $\forall f \in \mathcal{F}$, there exists $i \in [N]$ such that $\|f - f_i\| \leq \epsilon$. The covering number $N(\mathcal{F}, \|\cdot\|, \epsilon)$ is defined as the minimum cardinality $N$ of the covering set over all $\epsilon$-coverings of $\mathcal{F}$.*

It clear that $Reg(T)$ scales with $\sqrt{\log N(\mathcal{F}^*, \|\cdot\|, \epsilon)}$. Note that $N(\mathcal{F}^*, \|\cdot\|, \epsilon) \leq N(\mathcal{F}, \|\cdot\|, \epsilon)$ as $\mathcal{F}^* \subset \mathcal{F}$. The covering number shows clearly how extra causal bounds help improve the algorithm performance by shrinking the search space. Let $m = \inf_{w,a} l(w, a)$ and $M = \sup_{w,a} h(w, a)$. These bounds chip away the surface of the unit sphere and scoop out the concentric sphere of radius $m$. Therefore, the transfer learning algorithm only needs to search within a spherical shell with a thickness of at most $M - m$.

### A.7 Numerical setup

**Causal bounds.** To evaluate the effectiveness of our proposed Algorithm 5, we compare it with the method proposed by Li and Pearl (2022) when all variables are binary. Specifically, we randomly generate distributions $\mathbb{P}(a, y, w)$, as shown in Table 2, and set $\mathbb{P}(U = 1) = 0.1$. We implement

| $(A, Y, W)$ | $(0,0,0)$ | $(0,0,1)$ | $(0,1,0)$ | $(0,1,1)$ | $(1,0,0)$ | $(1,0,1)$ | $(1,1,0)$ | $(1,1,1)$ |
|---|---|---|---|---|---|---|---|---|
| $\mathbb{P}(a, y, w)$ | 0.2328 | 0.1784 | 0.1351 | 0.1467 | 0.0304 | 0.1183 | 0.0149 | 0.1433 |

Algorithm 5 with a batch size of 20000, and set $\epsilon = 0$ as Li and Pearl (2022) assume the given distributions are accurate.

| sample space for $x_{ijkl}$ | valid sample proportion |
|---|---|
| $[0, 1]$ | $\approx 0$ |
| $[\max\{0, \theta_{ijk} + \theta_l - 1\}, \min\{\theta_{ijk}, \theta_l\}]$ | $< 10^{-4}$ |
| support found by LP(3) | 0.3% |
| Algorithm 5 | 100% |

Table 5: Valid sample proportion with different sample spaces for the given example in Section 4.

**Transfer learning in MAB.** We perform simulation for 5-armed Bernoulli bandits with probability $0.1, 0.2, 0.3, 0.6, 0.8$. Simulations are partitioned into rounds of $T = 10^5$ trials averaged over 50 repetitions. For each task, we collect 1000 samples generated by a source agent and compute the empirical joint distribution. The estimated causal bounds without the knowledge of $F(u)$ (CUCB in Figure 2 (Zhang and Bareinboim, 2017)) are $h(a) = 0.9, 0.35, 0.92, 0.96, 0.92$, and $l(a) = 0, 0.08, 0.1, 0.3, 0.4$. The estimated causal bounds with the knowledge of $F(u)$ are $h(a) = 0.2, 0.25, 0.77, 0.7, 0.9$, and $l(a) = 0.01, 0.08, 0.19, 0.38, 0.71$.

**Transfer learning in contextual bandits.** We generate a function space $\mathcal{F} = \{(w - w_0)^\top (a - a_0)\}$ with a size of 50 by sampling parameters $w_0, a_0$ in $\mathbb{R}^d$ from $\mathcal{N}(0, 0.1)$, where $d = 10$. We then randomly choose a function as the true reward function $f^*$ from the first 5 functions, and generate the reward as $Y = f^*(W, A) + \mathcal{N}(0, 0.1)$, where the context $W$ is drawn i.d.d. from standard normal distributions and $A$ is the selected action. The whole action set $\mathcal{A}$ is randomly initialized from $[-1, 1]^d$ with a size of 10. We repeat each instance 50 times to obtain a smooth regret curve.

### A.8 Related work

Partially Observable Markov decision process (POMDP), including general partially observable dynamical systems (Uehara et al., 2022), also shares the similarity with our setting. Researchers have developed various methods to address causal inference in POMDPs. For example, Guo et al. (2022) use instrumental variables to identify causal effects, while Shi et al. (2022); Lu et al. (2023) extend this approach to general proxy variables in offline policy evaluation. In online reinforcement learning, Jin et al. (2019); Wang et al. (2021) use the backdoor criterion to explicitly adjust for confounding bias when confounders are fully observable. They also incorporate uncertainty from partially observable confounders into the Bellman equation and demonstrate provably optimal learning with linear function approximation in both fully and partially observable tasks. However, due to the complexity of reinforcement learning, transfer learning in POMDPs with the general function approximation still remains unknown. In our task 3, we address the problem of partially observable contextual bandit with the general function approximation under realizability assumption, which shows the potential to generalize to POMDPs and other related settings.

We also notice that our work is related with serval topics, including partial monitoring Cesa-Bianchi et al. (2006); Bartók and Szepesvári (2012); Lattimore and Szepesvári (2019). This line of work also shows how extra observations affect the regret upper bounds. Some papers Kirschner and Krause (2019) shows how one can achieve-near optimal regrets in linear contextual bandits with known context distribution. Their context distribution can be changed and chosen by the environment while we consider the fixed but partial observable context scenario. Hanna et al. (2022) shows how one can achieve near-optimal regrets in linear contextual bandit in a distributed setting. Though the learner cannot the observe the context, a central agent can transfer partial knowledge to help the leaner infer the contextual information.

Our work is closely related to several topics in the field. In particular, certain papers Kirschner and Krause (2019) have demonstrated how near-optimal regrets can be achieved in linear contextual bandits when the context distribution is known. These studies allow for the context distribution to be modified and chosen by the environment. However, our work focuses on a different scenario where the context is fixed but only partially observable. Furthermore, in the context of linear contextual bandits in a distributed setting, Hanna et al. (2022) presents a method for achieving near-optimal regrets. While the learner itself may not directly observe the context, a central agent is able to transfer partial knowledge to assist the learner in inferring the contextual information. Partial monitoring Cesa-Bianchi et al. (2006); Bartók and Szepesvári (2012); Lattimore and Szepesvári (2019) involves the influence of partial reward feedback. In contrast, we investigate the influence of partial contexts.

# B  DEFERRED PROOFS

## B.1  PROOF OF MENTIONED FACTS

**Fact B.1.1** *Given a series of known observational distributions $F^1, \cdots, F^n$, consider an optimization problem for causal effects:*

$$\inf / \sup CE(\mathcal{M})$$
$$F^i_{\mathcal{M}} = F^i, i = 1, \cdots, n,$$

*where $CE(\mathcal{M})$ is the desired casual effect and $F^i_{\mathcal{M}}$ is a distribution in the model $\mathcal{M}$. Here, the inf/sup is taken with respect to all compatible causal models $\mathcal{M}$. Then, a sufficient and necessary condition to identify $CE(\mathcal{M})$ is $LB = UB$, where $LB$ and $UB$ are the lower and upper bound solutions to the optimization problem.*

*Proof.* If $LB = HB$, then for any compatible model $\mathcal{M}_1$ and $\mathcal{M}_2$, we have

$$LB = CE(\mathcal{M}_1) = CE(\mathcal{M}_2) = UB.$$

According the definition of causal identification, the required causal effect $CE(\mathcal{M})$ can be fully identified.

On the contrary, suppose $CE(\mathcal{M})$ is causal identifiable. Then for any compatible model pair $\mathcal{M}_1$ and $\mathcal{M}_2$, we have $CE(\mathcal{M}_1) = CE(\mathcal{M}_2)$. Traveling over all compatible models immediately yields

$$LB = CE(\mathcal{M}_1) = CE(\mathcal{M}_2) = UB.$$

$\square$

**Fact B.1.2** *During discretization, equality constraints in Theorem A.1 are automatically satisfied in the sense of integration.*

*Proof.* The first constraint has been checked.

For the second equality, we integrate over $\mathcal{U}_l$ and have

$$\int_{u \in \mathcal{U}_l} dF(u) du$$
$$= \int_{a \in \mathcal{A}, y \in \mathcal{Y}, w \in \mathcal{W}, u \in \mathcal{U}_l} dF(a, y, w, u) du$$
$$= \sum_{ijk} \int_{a \in \mathcal{A}_i, \in \mathcal{Y}_j, w \in \mathcal{W}_k, u \in \mathcal{U}_l} dF(a, y, w, u) du$$
$$= \sum_{ijk} x_{ijkl}.$$

Hence, the second equality constraint holds in the sense of integration.

For the third and the fourth equality constrains, we can do integration over corresponding blocks and check the equality in the same way.

The conditional distribution in the fifth equality can be approximated by $x_{ijkl}$. See details in the proof of approximating objective $\hat{\mathbb{E}}_{\mathcal{M}}[Y|do(a)]$.

$\square$

**Fact B.1.3** *The object in Theorem A.1 after discretization is approximately equal to $\hat{\mathbb{E}}_{\mathcal{M}}[Y|do(a)]$.*

*Proof.* We use the average values to approximate distributions at certain points. First, we only need to consider the value

$$\int_{y\in\mathcal{Y}_j, w\in\mathcal{W}_k, u\in\mathcal{U}_l} y \frac{dF(a,y,w,u)dF(w,u)}{dF(a,w,u)},$$

as summing over $j, k, l$ can yield the object.

Suppose the given $a \in \mathcal{A}_i$ and let $vol(\cdot)$ denote the volume of the given block. For the values of distributions in $\mathcal{A}_i \times \mathcal{Y}_j \times \mathcal{W}_k \times \mathcal{U}_l$, we have

$$dF(a,y,w,u) \approx \frac{dadydwdu}{vol(\mathcal{A}_i)vol(\mathcal{Y}_j)vol(\mathcal{W}_k)vol(\mathcal{U}_l)} \int_{a\in\mathcal{A}_i, y\in\mathcal{Y}_j, w\in\mathcal{W}_k, u\in\mathcal{U}_l} dF(a,y,w,u)$$

$$= \frac{x_{ijkl}dadydwdu}{vol(\mathcal{A}_i)vol(\mathcal{Y}_j)vol(\mathcal{W}_k)vol(\mathcal{U}_l)},$$

$$dF(a,w,u) \approx \frac{dadwdu}{vol(\mathcal{A}_i)vol(\mathcal{W}_k)vol(\mathcal{U}_l)} \int_{a\in\mathcal{A}_i, w\in\mathcal{W}_k, u\in\mathcal{U}_l} dF(a,w,u)$$

$$= \frac{dadwdu}{vol(\mathcal{A}_i)vol(\mathcal{W}_k)vol(\mathcal{U}_l)} \int_{a\in\mathcal{A}_i, y\in\mathcal{Y}, w\in\mathcal{W}_k, u\in\mathcal{U}_l} dF(a,y,w,u)$$

$$= \frac{\sum_{j'} x_{ij'kl}dadwdu}{vol(\mathcal{A}_i)vol(\mathcal{W}_k)vol(\mathcal{U}_l)},$$

$$dF(w,u) \approx \frac{dwdu}{vol(\mathcal{W}_k)vol(\mathcal{U}_l)} \int_{w\in\mathcal{W}_k, u\in\mathcal{U}_l} dF(w,u)$$

$$= \frac{dwdu}{vol(\mathcal{W}_k)vol(\mathcal{U}_l)} \int_{a\in\mathcal{A}, y\in\mathcal{Y}, w\in\mathcal{W}_k, u\in\mathcal{U}_l} dF(a,y,w,u)$$

$$= \frac{\sum_{i',j'} x_{i'j'kl}dwdu}{vol(\mathcal{W}_k)vol(\mathcal{U}_l)}.$$

Plugging all above equalities yields

$$\int_{y\in\mathcal{Y}_j, w\in\mathcal{W}_k, u\in\mathcal{U}_l} y \frac{dF(a,y,w,u)dF(w,u)}{dF(a,w,u)}$$

$$\approx \frac{x_{ijkl}\sum_{i',j'} x_{i'j'kl}}{\sum_{j'} x_{ij'kl}} \int_{y\in\mathcal{Y}_j, w\in\mathcal{W}_k, u\in\mathcal{U}_l}$$

$$= \frac{x_{ijkl}\sum_{i',j'} x_{i'j'kl}}{\sum_{j'} x_{ij'kl}} \int_{y\in\mathcal{Y}_j, w\in\mathcal{W}_k, u\in\mathcal{U}_l} ydydydu/vol(\mathcal{Y}_j)vol(\mathcal{W}_k)vol(\mathcal{U}_l)$$

$$\approx \frac{y_j x_{ijkl}\sum_{i',j'} x_{i'j'kl}}{\sum_{j'} x_{ij'kl}}.$$

If $y_j$ is chosen to be $\frac{\int_{y\in\mathcal{Y}_j} ydy}{\int_{y\in\mathcal{Y}_j} dy}$, then the last symbol of approximation can be replaced with the symbol of equal. If $\mathcal{Y}_j$ is an interval, $y_j$ can be chosen as the midpoint of $\mathcal{Y}_j$. $\square$

The step of approximating $dF$ is crucial in reducing the approximation error. For absolutely continuous cumulative distribution functions, the approximation error will converge to zero as the diameter of each block approaches zero. Furthermore, if all random variables are discrete, the approximation

error can be exactly zero when using the natural discretization. In this case, the original objective can be expressed as

$$\sum_{y \in \mathcal{Y}, w \in \mathcal{W}, u \in \mathcal{U}} y \frac{\mathbb{P}(A = a, Y = y, W = w, U = u)\mathbb{P}(W = w, U = u)}{\mathbb{P}(A = a, W = w, U = u)}.$$

Our discretization method can be regarded as approximating probability mass functions.

**Fact B.1.4** *If the estimator $\hat{\theta}_x$ for $\theta_x \in [0, 1]$ has $n_x$ i.i.d. samples, then the variance of $\hat{\theta}_x$ is at most $\frac{1}{4n_x}$.*

*Proof.* Since $\theta_x^2 \leq \theta_x$, then

$$Var(\theta_x) = \mathbb{E}[\theta_x^2] - (\mathbb{E}[\theta_x])^2 \leq \mathbb{E}[\theta_x] - (\mathbb{E}[\theta_x])^2 \leq \frac{1}{4}.$$

Hence, we have

$$Var(\hat{\theta}_x) = \frac{1}{n_x} Var(\theta_x) \leq \frac{1}{4n_x}.$$

$\square$

## B.2    PROOF OF THEOREM A.1

*Proof.* Since $W$ and $U$ satisfies the back-door criterion, we can condition on $W, U$ to identify the causal effect $\mathbb{E}[Y|do(a_0)]$ We have

$$\mathbb{E}[Y|do(a_0)] = \int_{w \in \mathcal{W}, u \in \mathcal{U}} \mathbb{E}[Y|do(a_0), w, u] dF(w, u)$$

$$= \int_{w \in \mathcal{W}, u \in \mathcal{U}} \mathbb{E}[Y|a_0, w, u] dF(w, u)$$

$$= \int_{w \in \mathcal{W}, u \in \mathcal{U}} \int_{y \in \mathcal{Y}} y dF(y|a_0, w, u) dF(w, u).$$

The equalities come from the normalization properties of distribution functions. The inequalities come from the estimation error. $\square$

## B.3    PROOF OF CONVERGENCE RESULTS OF ALGORITHM 5

We first prove the following lemma to show that our sampling algorithm can cover all values in the feasible region $\mathcal{D}$. We denote the truncated distribution to $[l, h]$ from the user-given distribution $F_s$ when $x_i$ is given in sequential LPs as $F_s(x|x_i, [l, h])$.

**Lemma B.1** *The Algorithm 4 induces a distribution on the given simplex $\mathcal{D}$.*

*Proof.* We need to prove the sample generated by Algorithm 4 can exactly cover the region of $\mathcal{D}$. On the one hand, for any output $\mathbf{x}$, the feasibility of each component of $\mathbf{x}$ indicates that $\mathbf{x}$ must lie in $\mathcal{D}$. On the another hand, for any $\hat{\mathbf{x}} \in \mathcal{D}$, we show that this point can be generated by solving sequential LPs. Since $\hat{\mathbf{x}} \in \mathcal{D}$, $\hat{\mathbf{x}}$ is a feasible solution to the first LP

$$\min / \max x_1$$
$$s.t. A\mathbf{x} \leq \mathbf{b}, \mathbf{x} \geq \mathbf{0}.$$

One can check the feasibility of $\hat{\mathbf{x}}$ in the following LPs

$$\min / \max x_i$$
$$s.t. A\mathbf{x} \leq \mathbf{b}, \mathbf{x} \geq \mathbf{0}$$
$$x_1 = \hat{x}_1, \cdots, x_j = \hat{x}_j, j = 1, 2, \cdots, i - 1,$$

because $\hat{\mathbf{x}} \in \mathcal{D}$ and the previous $i$ components are exactly equal to those of $\hat{\mathbf{x}}$.

Suppose that the number of components of $\hat{\mathbf{x}}$ is $d$. Then the induced distribution is

$$F(\hat{\mathbf{x}}) = F_s(x_1|[l_1, h_1]) \prod_{i=2}^{d} F_s(x_i|[l_i, h_i], x_j, j = 1, 2, \cdots, i-1).$$

$\square$

We now give the proof for Proposition B.1.

**Proposition B.1** *Assume that the sampling measure* $\mathbb{P}_s$ *satisfies* $\forall \mathbf{x} \in \mathcal{D}$, *and* $\forall \delta > 0$,

$$\mathbb{P}_s(\mathcal{B}(\mathbf{x}, \delta) \cap \mathcal{D}) > 0,$$

*where* $\mathcal{B}(\mathbf{x}, \delta)$ *is a ball centered at* $\mathbf{x}$ *with radius* $\delta$. *If the discrepancy parameter is set to* $0$, *then* $b_{(1)}(a)$ *converges to* $\hat{l}(a)$ *in probability and* $b_{(B)}(a)$ *converges to* $\hat{h}(a)$ *in probability for* $B \to \infty$.

*Proof.* The discretization optimization problem (5) has one-to-one correspondence between $\mathbf{x}$ and each causal model where all random variables are discrete. From Lemma B.1, we know the one-to-one correspondence between each model and $\mathbf{x}$. Therefore, $[\hat{l}(a), \hat{h}(a)]$ is the support of the induced distribution on casual bounds.

As shown in $\hat{\mathbb{E}}_{\mathcal{M}}[Y|do(a)]$, the object can be regarded as a function of $\mathbf{x}$. We define $\phi = \hat{\mathbb{E}}_{\mathcal{M}}[Y|do(a)]$ which is a continuous mapping from $\mathcal{D}$ to $[0, 1]$. The continuity of $\phi$ is clear for $\sum j' x_{ij'kl} > 0$. When $\sum_{j'} x_{ij'kl} = 0$, the non-negativity of $x_{ijkl}$ implies $\sum_{i',j'} x_{i'j'kl} = 0$. In this case, we can set the value of $\frac{y_j x_{ijkl} \sum_{i',j'} x_{i'j'kl}}{\sum_{j'} x_{ij'kl}}$ to be 0 to maintain continuity.

Given that $\mathcal{D}$ is a compact set, there exists $\mathbf{x}_l$ such that $\phi(\mathbf{x}_l) = \hat{l}(a)$. The continuity indicates that $\forall \epsilon > 0$, there exists $\delta > 0$ such that $\phi(\mathbf{x}) < \hat{l}(a) + \epsilon$ for all $\mathbf{x} \in \mathcal{B}(x_h, \delta)$, then

$$\mathbb{P}(b(a) < \hat{l}(a) + \epsilon) = \mathbb{P}_s \left( \bigcup_{b < \hat{l}(a)+\epsilon} \{\mathbf{x} \in \mathcal{D}|\phi(\mathbf{x}) = b\} \right) \geq \mathbb{P}_s(\mathcal{B}(\mathbf{x}_h, \delta)) > 0.$$

This implies that

$$\mathbb{P}(b_{(1)}(a) < \hat{l}(a) + \epsilon) = 1 - (1 - \mathbb{P}(b(a) < \hat{l}(a) + \epsilon))^B \to 1$$

as $B \to \infty$. Since $b_{(1)}(a)$ is a feasible solution to the discrete optimization problem, we have

$$\mathbb{P}(\hat{l}(a) \leq b_{(1)}(a) < \hat{l}(a) + \epsilon) = 1 - (1 - \mathbb{P}(b(a) < \hat{l}(a) + \epsilon))^B \to 1$$

which implies $b_{(1)}(a) \to \hat{l}(a)$ in probability.

Similarly, we can prove that $\mathbb{P}(b_{(B)}(a) > \hat{h}(a) - \epsilon) < 1$ and thus $b_{(B)}(a) \to \hat{h}(a)$ in probability.

$\square$

*Proof.* The one-to-one correspondence between $\mathbf{x}$ and each causal model has been proved in Lemma B.1. Therefore, $[\hat{l}(a), \hat{h}(a)]$ is the support of the induced distribution on casual bounds. As shown in the proof of Proposition B.1, the defined $\phi = \hat{\mathbb{E}}_{\mathcal{M}}[Y|do(a)]$ is a continuous mapping from $\mathcal{D}$ to $[0, 1]$.

Given that $\mathcal{D}$ is a compact set, there exists $\mathbf{x}_l$ such that $\phi(\mathbf{x}_l) = \hat{l}(a)$. The property of **OPT** implies that there exists $\delta > 0$ such that

$$\mathbf{OPT}(min/max, \hat{\mathbb{E}}_{\mathcal{M}}[Y|do(a)], \mathcal{D}, \mathbf{x}) = \mathbf{x}_l, \forall \mathbf{x} \in \mathcal{B}(\mathbf{x}_l, \delta) \cap \mathcal{D}.$$

Hence, we have

$$\mathbb{P}(b(a) = \hat{l}(a)) \geq \mathbb{P}_s(\mathcal{B}(\mathbf{x}_l, \delta) \cap \mathcal{D}) > 0.$$

Due to Borel-Cantelli lemma we can prove that $b_{(1)}(a) \to \hat{l}(a)$ almost surely, because each $b_i(a)$ is independently sampled by Algorithm 1. Similarly, we can prove that $b_{(B)}(a) \to \hat{h}(a)$ almost surely.

$\square$

### B.4 PROOF OF REGRETS IN MAB

We first prove Theorem A.2.

*Proof.* **Case 1**: $h(a) < \max_{i \in \mathcal{A}} l(i)$

From the algorithmic construction, we know that such arm $a$ is removed and thus

$$\mathbb{E}[N_a(T)] = 0.$$

**Case 2**: $\max_{i \in \mathcal{A}} l(i) \leq h(a) < \mu^*$.

Let $a^* = \arg\max_{a \in \mathcal{A}} \mathbb{E}[Y|do(a)]$ be the optimal action with respect to $w$. Define the following event

$$\mathcal{E}(t) = \left\{ \hat{\mu}_a \in [\mu_a - \frac{\log t}{n_a(t)}, \mu_a + \frac{\log t}{n_a(t)}], \forall a \in \mathcal{A} \right\}$$

Then the Chernoff's bound yields

$$\mathbb{P}(\overline{\mathcal{E}(t)}) \leq \sum_{a \in \mathcal{A}} \exp(-2n_a(t) \times \frac{\log t}{n_a(t)}) \leq \frac{|\mathcal{A}|}{t^2}$$

For any given $w$, the event $\{A_t = a\}$ implies $\hat{U}_a(t) > \hat{U}_{a^*}(t)$. However,

$$\mu^* > h(a) \geq \hat{U}_a(t)$$

and

$$\hat{U}_{a^*}(t) \geq \mu^*$$

if $\mathcal{E}(t)$ holds. This leads to contradiction. Therefore,

$$\begin{aligned}
\mathbb{E}[N_a(T)] &= \sum_{t=1}^{T} \mathbb{P}(A_t = a) \\
&= \sum_{t=1}^{T} \mathbb{P}(A_t = a|\mathcal{E}(t))\mathbb{P}(\mathcal{E}(t)) + \mathbb{P}(A_t = a|\overline{\mathcal{E}(t)})\mathbb{P}(\overline{\mathcal{E}(t)}) \\
&\leq \sum_{t=1}^{T} \mathbb{P}(\overline{\mathcal{E}(t)}) \\
&\leq \sum_{t=1}^{T} \frac{|\mathcal{A}|}{t^2} \\
&\leq \frac{|\mathcal{A}|\pi^2}{6}.
\end{aligned}$$

**Case 3**: $h(a) \geq \mu^*$

We reuse the notation $\mathcal{E}(t)$ in the case 2. Condition on the event $\bigcap_{t=1}^{T} \mathcal{E}(t)$, if $n_a(t) \geq \frac{8\log T}{\Delta_a^2}$, then

$$\hat{U}_a(t) \leq U_a(t) = \mu_a + \sqrt{\frac{2\log t}{n_a(t)}} \leq \mu_a + \frac{1}{2}\Delta_a = \mu^* \leq \hat{U}_{a^*}(t),$$

so Algorithm 2 will not choose the action $a$ at the round $t$. Therefore,

$$\mathbb{E}[N_a(T)] \leq \mathbb{E}\left[ N_a(T) \Big| \bigcap_{t=1}^{T} \mathcal{E}(t) \right] + T\mathbb{P}\left( \overline{\bigcap_{t=1}^{T} \mathcal{E}(t)} \right) \leq \frac{8\log T}{\Delta_a^2} + T\mathbb{P}\left( \overline{\bigcup_{t=1}^{T} \mathcal{E}(t)} \right).$$

We conclude the proof by showing

$$T\mathbb{P}\left( \overline{\bigcup_{t=1}^{T} \mathcal{E}(t)} \right) \leq T\sum_{t=1}^{T} \frac{|\mathcal{A}|}{t^2} < T \times \frac{|\mathcal{A}|}{T} = |\mathcal{A}|.$$

$\square$

Actually, the proof is just a simple modification of that in Theorem 3.1, because MAB can be regarded as a special case of contextual bandits.

**Theorem B.1** *Consider a MAB bandit problem with $|\mathcal{A}| < \infty$. Denote*

$$\widetilde{\mathcal{A}^*} = \mathcal{A} - \{a \in \mathcal{A} | h(a) < \mu^*\}.$$

*Then the regret of Algorithm 6 is upper bounded by*

$$\mathbb{E}[Reg(T)] \leq \sqrt{8(|\widetilde{\mathcal{A}^*}(w)| - 1)T \log T}.$$

*Proof.* Theorem A.2 shows that

$$\begin{aligned}
\mathbb{E}[Reg(T)] &= \sum_{a \in \mathcal{A}} \Delta_a \mathbb{E}[N_a(T)] \\
&\leq \sum_{a \in \widetilde{\mathcal{A}^*}} \frac{8 \log T}{\Delta_a} \mathbf{I}\{\Delta_a \geq \Delta\} + T\Delta + \mathcal{O}(|\mathcal{A}|) \\
&\leq \frac{8(|\widetilde{\mathcal{A}^*}| - 1) \log T}{\Delta} + T\Delta + \mathcal{O}(|\mathcal{A}|).
\end{aligned}$$

Specifying $\Delta = \sqrt{\frac{8(|\widetilde{\mathcal{A}^*}| - 1) \log T}{T}}$ concludes the proof. $\qquad\square$

Denote the contextual bandit instances with prior knowledge $l(a)$ and $h(a)$ as

$$\mathfrak{M} = \{\text{MAB bandit instances with } l(a) \leq \mu_a \leq h(a), \forall a \in \mathcal{A}\}.$$

**Theorem B.2** *Suppose $|\mathcal{A}| < \infty$. Then for any algorithm $\mathsf{A}$, there exists an absolute constant $c > 0$ such that*

$$\min_{\mathsf{A}} \sup_{\mathfrak{M}} Reg(T) \geq \frac{1}{27}\sqrt{(|\widetilde{\mathcal{A}^*}| - 1)T}.$$

*Proof.* This is a direct corollary of MAB regret lower bound, because any arm in $\widetilde{\mathcal{A}^*}$ cannot be the optimal one. $\qquad\square$

### B.5 OMITTED THEOREMS IN TASK 3

From the rule of do-calculus, we have

$$\begin{aligned}
\mathbb{E}[Y|do(a), w] &= \int_{u \in \mathcal{U}} \mathbb{E}[Y|do(a), w, u] dF(u) \\
&= \int_{u \in \mathcal{U}} \mathbb{E}[Y|a, w, u] dF(u).
\end{aligned}$$

The last equality is due to the rule of do-calculus as $W$ and $U$ is sufficient to block all back-door paths from $A$ to $Y$.

**Theorem B.3** *Given a causal diagram $\mathcal{G}$ and a distribution compatible with $\mathcal{G}$, let $\{W, U\}$ be a set of variables satisfying the back-door criterion in $\mathcal{G}$ relative to an ordered pair $(A, Y)$, where $\{W, U\}$ is partially observable, i.e., only probabilities $\hat{F}(a, y, w)$ and $\hat{F}(u)$ with the maximum estimation error $\epsilon$, the causal effects of $A = a_0$ on $Y$ when $W = w_0$ occurs are then bounded as follows:*

$$l(w_0, a_0) \leq \mathbb{E}[Y|do(a_0), w_0] \leq h(w_0, a_0),$$

*where $l(w_0, a_0)$ and $h(w_0, a_0)$ are solutions to the following functional optimization problem for any given $a_0$ and $w_0$*

$$l(w_0, a_0) = \inf \int_{w \in \mathcal{W}, u \in \mathcal{U}} \int_{y \in \mathcal{Y}} y dF(y|a_0, w_0, u) dF(u)$$

$$h(w_0, a_0) = \sup \int_{w \in \mathcal{W}, u \in \mathcal{U}} \int_{y \in \mathcal{Y}} y dF(y|a_0, w_0, u) dF(u)$$

$$s.t. F(y|a, w, u)F(a, w, u) = F(a, y, w, u), \forall (a, y, w, u) \in \mathcal{A} \times \mathcal{Y} \times \mathcal{W} \times \mathcal{U}$$

$$\int_{y \in \mathcal{Y}} dF(a, y, w, u) = F(a, w, u), \forall (a, w, u) \in \mathcal{A} \times \mathcal{W} \times \mathcal{U}$$

$$\int_{a \in \mathcal{A}, y \in \mathcal{Y}, w \in \mathcal{W}} dF(a, y, w, u) = F(u), \forall u \in \mathcal{U}$$

$$\int_{u \in \mathcal{U}} dF(a, y, w, u) = F(a, y, w), \forall (a, y, w) \in \mathcal{A} \times \mathcal{W} \times \mathcal{U}$$

$$|F(a, y, w) - \hat{F}(a, y, w)| \le \epsilon, \forall (a, y, w) \in \mathcal{A} \times \mathcal{Y} \times \mathcal{W}$$

$$|F(u) - \hat{F}(u)| \le \epsilon, \forall u \in \mathcal{U}.$$

*Here the inf/sup is taken with respect to all unknown cumulative distribution functions $F(a, y, w, u)$, $F(a, y, w)$, $F(a, w, u)$, $F(u)$.*

*Proof.* The object is shown at the beginning of this subsection. The equalities come from the normalization properties, and inequalities follow from estimation error. □

Denote the following optimization problem

$$\max / \min \hat{\mathbb{E}}_{\mathcal{M}}[Y|do(a), w]$$
$$\sum_l x_{ijkl} = \theta_{ijk}, \forall i \in [n_{\mathcal{A}}], j \in [n_{\mathcal{Y}}], k \in [n_{\mathcal{W}}] \tag{8}$$
$$\sum_{ijk} x_{ijkl} = \theta_l, \forall l \in [n_{\mathcal{U}}].$$

where the objective $\hat{\mathbb{E}}_{\mathcal{M}}[Y|do(a), w]$ after discretization is defined as

$$\hat{\mathbb{E}}_{\mathcal{M}}[Y|do(a), w] = \sum_{jkl} \frac{y_j \theta_l x_{ijkl}}{\sum_{j'} x_{ij'kl}}. \tag{9}$$

Denote the solutions to (8) as $\hat{l}(w, a)$ and $\hat{h}(w, a)$. Note that the optimization problem (7) shares the same feasible region with that of (1).

**Proposition B.2** *Assume that the sampling measure $\mathbb{P}_s$ satisfies $\forall \mathbf{x} \in \mathcal{D}$, and $\forall \delta > 0$,*

$$\mathbb{P}_s(\mathcal{B}(\mathbf{x}, \delta) \cap \mathcal{D}) > 0,$$

*where $\mathcal{B}(\mathbf{x}, \delta)$ is a ball centered at $\mathbf{x}$ with radius $\delta$. If the discrepancy parameter is set to $0$, then $b_{(1)}(w, a)$ converges to $\hat{l}(w, a)$ in probability and $b_{(B)}(w, a)$ converges to $\hat{h}(w, a)$ in probability for any given $(w, a)$ and $B \to \infty$.*

*Proof.* As shown in $\hat{\mathbb{E}}_{\mathcal{M}}[Y|do(a), w]$, the object can also be regarded as a function of $\mathbf{x}$. We define $\phi = \hat{\mathbb{E}}_{\mathcal{M}}[Y|do(a), w]$ which is a continuous mapping from $\mathcal{D}$ to $[0, 1]$. The continuity of $\phi$ holds similarly.

Given that $\mathcal{D}$ is a compact set, there exists $\mathbf{x}_l$ such that $\phi(\mathbf{x}_l) = \hat{l}(w, a)$. The continuity indicates that $\forall \epsilon > 0$, there exists $\delta > 0$ such that $\phi(\mathbf{x}) < \hat{l}(w, a) + \epsilon$ for all $\mathbf{x} \in \mathcal{B}(x_h, \delta)$, then

$$\mathbb{P}(b(w, a) < \hat{l}(w, a) + \epsilon) = \mathbb{P}_s \left( \bigcup_{b < \hat{l}(w, a) + \epsilon} \{\mathbf{x} \in \mathcal{D}|\phi(\mathbf{x}) = b\} \right) \ge \mathbb{P}_s(\mathcal{B}(\mathbf{x}_h, \delta)) > 0.$$

This implies that

$$\mathbb{P}(b_{(1)}(w,a) < \hat{l}(w,a) + \epsilon) = 1 - (1 - \mathbb{P}(b(w,a) < \hat{l}(w,a) + \epsilon))^B \to 1$$

as $B \to \infty$. Since $b_{(1)}(a)$ is a feasible solution to the discrete optimization problem, we have

$$\mathbb{P}(\hat{l}(w,a) \le b_{(1)}(w,a) < \hat{l}(w,a) + \epsilon) = 1 - (1 - \mathbb{P}(b(w,a) < \hat{l}(w,a) + \epsilon))^B \to 1$$

which implies $b_{(1)}(w,a) \to \hat{l}(w,a)$ in probability.

Similarly, we can prove that $\mathbb{P}(b_{(B)}(w,a) > \hat{h}(w,a) - \epsilon) < 1$ and thus $b_{(B)}(w,a) \to \hat{h}(w,a)$ in probability.

$\square$

We can also incorporate the optimization procedure **OPT**. Replacing the objective $\hat{\mathbb{E}}_{\mathcal{M}}[Y|do(a)]$ in Algorithm 1 with $\hat{\mathbb{E}}_{\mathcal{M}}[Y|do(a), w]$, we can also prove the similar almost surely convergence result.

**Proposition B.3** *Assume that the sampling measure $\mathbb{P}_s$ satisfies $\forall \mathbf{x} \in \mathcal{D}$, and $\forall \delta > 0$,*

$$\mathbb{P}_s(\mathcal{B}(\mathbf{x}, \delta) \cap \mathcal{D}) > 0,$$

*where $\mathcal{B}(\mathbf{x}, \delta)$ is a ball centered at $\mathbf{x}$ with radius $\delta$. Given a deterministic procedure **OPT** which satisfies for any local optima $\mathbf{x}_{loc}$, there exists $\delta > 0$ such that for any initial guess $\mathbf{x}_0 \in \mathcal{B}(\mathbf{x}_{loc}, \delta) \cap \mathcal{D}$, **OPT** can output $\mathbf{x}_{loc}$ as a result. If the discrepancy parameter is set to $0$, then $b_{(1)}(w,a)$ and $b_{(B)}(w,a)$ converge almost surely to $\hat{l}(w,a)$ and $\hat{h}(w,a)$ for $B \to \infty$, respectively.*

*Proof.* The one-to-one correspondence between $\mathbf{x}$ and each causal model has been proved in Lemma B.1. Therefore, $[\hat{l}(w,a), \hat{h}(w,a)]$ is the support of the induced distribution on casual bounds. As shown in the proof of Proposition B.1, the defined $\phi = \hat{\mathbb{E}}_{\mathcal{M}}[Y|do(a), w]$ is a continuous mapping from $\mathcal{D}$ to $[0, 1]$.

Given that $\mathcal{D}$ is a compact set, there exists $\mathbf{x}_l$ such that $\phi(\mathbf{x}_l) = \hat{l}(w,a)$. The property of **OPT** implies that there exists $\delta > 0$ such that

$$\mathbf{OPT}(min/max, \hat{\mathbb{E}}_{\mathcal{M}}[Y|do(a)], \mathcal{D}, \mathbf{x}) = \mathbf{x}_l, \forall \mathbf{x} \in \mathcal{B}(\mathbf{x}_l, \delta) \cap \mathcal{D}.$$

Hence, we have

$$\mathbb{P}(b(w,a) = \hat{l}(w,a)) \ge \mathbb{P}_s(\mathcal{B}(\mathbf{x}_l, \delta) \cap \mathcal{D}) > 0.$$

Due to Borel-Cantelli lemma we can prove that $b_{(1)}(w,a) \to \hat{l}(w,a)$ almost surely, because each $b_i(w,a)$ is independently sampled by Algorithm 1. Similarly, we can prove that $b_{(B)}(w,a) \to \hat{h}(w,a)$ almost surely.

$\square$

## B.6 Proof of Theorem 3.1

*Proof.* We consider any given $w$ in the following proof.

**Case 1**: $h(w,a) < \max_{i \in \mathcal{A}} l(w,i)$

From the algorithmic construction, we know that such arm $a$ is removed and thus

$$\mathbb{E}[N_a(T_w)] = 0.$$

**Case 2**: $\max_{i \in \mathcal{A}} l(w,i) \le h(w,a) < \mu_w^*$.

Let $a_w^* = \arg\max_{a \in \mathcal{A}} \mathbb{E}[Y|do(a), w]$ be the optimal action with respect to $w$. Define the following event

$$\mathcal{E}_w(t) = \left\{ \hat{\mu}_{w,a} \in [\mu_{w,a} - \frac{\log t}{n_{w,a}(t)}, \mu_{w,a} + \frac{\log t}{n_{w,a}(t)}], \forall a \in \mathcal{A} \right\}$$

Then the Chernoff's bound yields

$$\mathbb{P}(\overline{\mathcal{E}_w(t)}) \le \sum_{a \in \mathcal{A}} \exp(-2n_{w,a}(t) \times \frac{\log t}{n_{w,a}(t)}) \le \frac{|\mathcal{A}|}{t^2}$$

For any given $w$, the event $\{A_t = a\}$ implies $\hat{U}_{w,a}(t) > \hat{U}_{w,a_w^*}(t)$. However,

$$\mu_w^* > h(w,a) \geq \hat{U}_{w,a}(t)$$

and

$$\hat{U}_{w,a_w^*}(t) \geq \mu_w^*$$

if $\mathcal{E}_w(t)$ holds. This leads to contradiction. Therefore,

$$\begin{aligned}
\mathbb{E}[N_a(T_w)] &= \sum_{t=1}^{T_w} \mathbb{P}(A_t = a | w_t = w) \\
&= \sum_{t=1}^{T_w} \mathbb{P}(A_t = a | w_t = w, \mathcal{E}_w(t)) \mathbb{P}(\mathcal{E}_w(t)) + \mathbb{P}(A_t = a | w_t = w, \overline{\mathcal{E}_w(t)}) \mathbb{P}(\overline{\mathcal{E}_w(t)}) \\
&\leq \sum_{t=1}^{T_w} \mathbb{P}(\overline{\mathcal{E}_w(t)}) \\
&\leq \sum_{t=1}^{T_w} \frac{|\mathcal{A}|}{t^2} \\
&\leq \frac{|\mathcal{A}|\pi^2}{6}.
\end{aligned}$$

**Case 3**: $h(w,a) \geq \mu_w^*$

We reuse the notation $\mathcal{E}_w(t)$ in the case 2. Condition on the event $\bigcap_{t=1}^{T_w} \mathcal{E}_w(t)$, if $n_{w,a}(t) \geq \frac{8 \log T_w}{\Delta_{w,a}^2}$, then

$$\hat{U}_{w,a}(t) \leq U_{w,a}(t) = \mu_{w,a} + \sqrt{\frac{2 \log t}{n_{w,a}(t)}} \leq \mu_{w,a} + \frac{1}{2}\Delta_{w,a} = \mu_w^* \leq \hat{U}_{w,a_w^*}(t),$$

so Algorithm 2 will not choose the action $a$ at the round $t$. Therefore,

$$\mathbb{E}[N_a(T_w)] \leq \mathbb{E}\left[N_a(T_w), \bigcap_{t=1}^{T_w} \mathcal{E}_w(t)\right] + T_w \mathbb{P}\left(\overline{\bigcap_{t=1}^{T_w} \mathcal{E}_w(t)}\right) \leq \frac{8 \log T_w}{\Delta_{w,a}^2} + T_w \mathbb{P}\left(\overline{\bigcup_{t=1}^{T_w} \mathcal{E}_w(t)}\right).$$

We conclude the proof by showing

$$T_w \mathbb{P}\left(\overline{\bigcup_{t=1}^{T_w} \mathcal{E}_w(t)}\right) \leq T_w \sum_{t=1}^{T_w} \frac{|\mathcal{A}|}{t^2} < T_w \times \frac{|\mathcal{A}|}{T_w} = |\mathcal{A}|.$$

$\square$

### B.7 PROOF OF THEOREM A.3

*Proof.* Let $\Delta_w$ be the constant with respect to $w$ that we will specify later. From the proof of Theorem 3.1, we know that the expected regret can be upper bounded as

$$\begin{aligned}
\mathbb{E}[Reg(T)] &= \sum_{w \in \mathcal{W}} \sum_{a \in \mathcal{A}} \Delta_{w,a} \mathbb{E}[N_a(T_w)] \\
&\leq \sum_{w \in \mathcal{W}} \left( \sum_{a \in \widetilde{\mathcal{A}}^*(w)} \frac{8 \log T_w}{\Delta_{w,a}} \mathbf{I}\{\Delta_{w,a} \geq \Delta_w\} + T\Delta_w \right) + \mathcal{O}(|\mathcal{A}|) \\
&\leq \sum_{w \in \mathcal{W}} \left( \frac{8(|\widetilde{\mathcal{A}}^*(w)| - 1) \log T}{\Delta_w} + T_w \Delta_w \right) + \mathcal{O}(|\mathcal{A}|).
\end{aligned}$$

We select $\Delta_w = \sqrt{\frac{8|\widetilde{\mathcal{A}^*}(w)|\log T}{T_w}}$ so

$$\mathbb{E}[Reg(T)] \le \sum_{w \in \mathcal{W}} \sqrt{8(|\widetilde{\mathcal{A}^*}(w)| - 1)T_w \log T}.$$

By strong law of large numbers, we have

$$\liminf_{T \to \infty} \frac{\mathbb{E}[Reg(T)]}{\sqrt{T \log T}} \le \sum_{w \in \mathcal{W}} \sqrt{8(|\widetilde{\mathcal{A}^*}(w)| - 1)\liminf_{T \to \infty} \frac{T_w}{T}}$$

$$= \sum_{w \in \mathcal{W}} \sqrt{8(|\widetilde{\mathcal{A}^*}(w)| - 1)\mathbb{P}(W = w)}.$$

$\square$

*Proof.* Consider $|\mathcal{W}|$ MAB instances. For any given context $w$, the set that the optimal arm will be in is $\mathcal{A}^*(w)$. For any algorithm A, let $\mathsf{A}_w$ be the induced algorithm of A when $w$ occurs. From the minimax theorem for MAB instances (Lattimore and Szepesvári, 2020), we know that there exists a MAB instance for each $w$ such that the regret of $\mathsf{A}_w$ is at least $\frac{1}{27}\sqrt{(|\widetilde{\mathcal{A}^*}(w)| - 1)T_w}$, where $T_w$ is the number of occurrence of $w$. Hence,

$$Reg(T) \ge \sum_{w \in \mathcal{W}} \frac{1}{27}\sqrt{(|\widetilde{\mathcal{A}^*}(w)| - 1)T_w}.$$

and almost surely,

$$\liminf_{T \to \infty} \frac{Reg(T)}{\sqrt{T}} \ge \frac{1}{27} \sum_{w \in \mathcal{W}} \sqrt{(|\widetilde{\mathcal{A}^*}(w)| - 1) \cdot \liminf_{T \to \infty} \frac{T_w}{T}}$$

$$= \frac{1}{27} \sum_{w \in \mathcal{W}} \sqrt{(|\widetilde{\mathcal{A}^*}(w)| - 1)\mathbb{P}(W = w)}.$$

$\square$

## B.8 PROOF OF THEOREM 3.3

The framework presented in (Simchi-Levi and Xu, 2021; Foster et al., 2020) provides a method to analyze contextual bandit algorithms in the universal policy space $\Psi$. In this paper, we mainly focus on a subspace of $\Psi$ shaped by causal bounds. We demonstrate that the action distribution $p_m$ selected in Algorithm 3 possesses desirable properties that contribute to achieving low regrets.

For each epoch $m$ and any round $t$ in epoch $m$, for any possible realization of $\gamma_t$, $\hat{f}_m$, we define the universal policy space of $\Psi$:

$$\Psi = \prod_{w \in \mathcal{W}} \mathcal{A}^*(w).$$

With abuse of notations, we define

$$\mathcal{R}(\pi) = \mathbb{E}_W[f^*(W, \pi(W))] \text{ and } Reg(\pi) = \mathcal{R}(\pi_{f^*}) - \mathcal{R}(\pi).$$

The above quantities do not depend on specific values of $W$. The following empirical version of above quantities are defined as

$$\widehat{\mathcal{R}}_t(\pi) = \hat{f}_{m(t)}(w, \pi(w)) \text{ and } \widehat{Reg}_t(\pi) = \mathbb{E}_W[\widehat{\mathcal{R}}_t(\pi_{\hat{f}_{m(t)}}) - \widehat{\mathcal{R}}_t(\pi)],$$

where $m(t)$ is the epoch of the round $t$.

Let $Q_m(\cdot)$ be the equivalent policy distribution for $p_m(\cdot|\cdot)$, i.e.,

$$Q_m(\pi) = \prod_{w \in \mathcal{W}} p_m(\pi(w)|w), \forall \pi \in \Psi.$$

The existence and uniqueness of such measure $Q_m(\cdot)$ is a corollary of Kolmogorov's extension theorem. Note that both $\Psi$ and $Q_m(\cdot)$ are $\mathcal{H}_{\tau_{m-1}}$-measurable, where $\mathcal{H}_t$ is the filtration up to the time $t$. We refer to Section 3.2 of (Simchi-Levi and Xu, 2021) for more detailed intuition for $Q_m(\cdot)$ and proof of existence. By Lemma 4 of (Simchi-Levi and Xu, 2021), we know that for all epoch $m$ and all rounds $t$ in epoch $m$, we can rewrite the expected regret in terms of our notations as

$$\mathbb{E}[Reg(T)] = \sum_{\pi \in \Psi} Q_m(\pi) Reg(\pi).$$

For simplicity, we define an epoch-dependent quantities

$$\rho_1 = 1, \rho_m = \sqrt{\frac{\eta \tau_{m-1}}{\log(2\delta^{-1}|\mathcal{F}^*| \log T)}}, m \geq 2,$$

so $\gamma_t = \sqrt{|\mathcal{A}^*(w_t)|} \rho_{m(t)}$ for $m(t) \geq 2$.

**Lemma B.2** *(Implicit Optimization Problem). For all epoch $m$ and all rounds $t$ in epoch $m$, $Q_m$ is a feasible solution to the following implicit optimization problem:*

$$\sum_{\pi \in \Psi} Q_m(\pi) \widehat{Reg}_t(\pi) \leq \mathbb{E}_W[\sqrt{|\mathcal{A}^*(W)|}]/\rho_m \tag{10}$$

$$\mathbb{E}_W \left[ \frac{1}{p_m(\pi(W)|W)} \right] \leq \mathbb{E}_W[\mathcal{A}^*(W)] + \mathbb{E}_W[\sqrt{|\mathcal{A}^*(W)|}] \rho_m \widehat{Reg}_t(\pi), \forall \pi \in \Psi. \tag{11}$$

*Proof.* Let $m$ and $t$ in epoch $m$ be fixed. Denote $\mathcal{P}(\mathcal{W})$ as the context distribution. We have

$$\sum_{\pi \in \Psi} Q_m(\pi) \widehat{Reg}_t(\pi)$$

$$= \sum_{\pi \in \Psi} Q_m(\pi) \mathbb{E}_{w_t} \left[ (\hat{f}_m(w_t, \pi_{\hat{f}_m}(w_t)) - \hat{f}_m(w_t, \pi(w_t))) \right]$$

$$= \mathbb{E}_{w_t \sim \mathcal{P}(\mathcal{W})} \left[ \sum_{a \in \mathcal{A}^*(w_t)} \sum_{\pi \in \Psi} \mathbf{I}\{\pi(w_t) = a\} Q_m(\pi)(\hat{f}_m(w_t, \pi_{\hat{f}_m}(w_t)) - \hat{f}_m(w_t, a)) \right]$$

$$= \mathbb{E}_{w_t \sim \mathcal{P}(\mathcal{W})} \left[ \sum_{a \in \mathcal{A}^*(w_t)} p_m(a|w_t)(\hat{f}_m(w_t, \pi_{\hat{f}_m}(w_t)) - \hat{f}_m(w_t, a)) \right].$$

The first and second equalities are the definitions of $\widehat{Reg}_t(\pi)$ and $Q_m(\pi)$, respectively.

Now for the context $w_t$, we have

$$\sum_{a \in \mathcal{A}^*(w_t)} p_m(a|w)(\hat{f}_m(w_t, \pi_{\hat{f}_m}(w_t)) - \hat{f}_m(w_t, a))$$

$$= \sum_{a \in \mathcal{A}^*(w_t) - \{\pi_{\hat{f}_m}(w_t)\}} \frac{\hat{f}_m(w_t, \pi_{\hat{f}_m}(w_t)) - \hat{f}_m(w_t, a)}{|\mathcal{A}^*(w_t)| + \gamma_t(\hat{f}_m(w_t, \pi_{\hat{f}_m}(w_t)) - \hat{f}_m(w_t, a))}$$

$$\leq [|\mathcal{A}^*(w_t)| - 1]/\gamma_t$$

$$\leq \sqrt{|\mathcal{A}^*(w_t)|}/\rho_m.$$

We plug in the above term and apply the i.d.d. assumption on $w_t$ to conclude the proof of the first inequality.

For the second inequality, we first observe that for any policy $\pi \in \Psi$, given any context $w \in \mathcal{W}$,

$$\frac{1}{p_m(\pi(w)|w)} = |\mathcal{A}^*(w)| + \gamma_t(\hat{f}_m(w, \pi_{\hat{f}_m}(w)) - \hat{f}_m(w, a)),$$

if $a \neq \pi_{\hat{f}_m}(w)$, and

$$\frac{1}{p_m(\pi(w)|w)} \leq \frac{1}{1/|\mathcal{A}^*(w)|} = |\mathcal{A}^*(w)| + \gamma_t(\hat{f}_m(w, \pi_{\hat{f}_m}(w)) - \hat{f}_m(w, a)),$$

if $a = \pi_{\hat{f}_m}(w)$. The result follows immediately by taking expectation over $w$. $\qquad\square$

Compared with IOP in (Simchi-Levi and Xu, 2021), the key different part is that $\mathbb{E}_W[|\mathcal{A}^*(W)|]$ is replaced by the cardinality $|\mathcal{A}|$ of the whole action set. Another different part is the universal policy space $\Psi$. We define $\Psi$ as $\prod_{w \in \mathcal{W}} \mathcal{A}^*(w)$ rather than $\prod_{w \in \mathcal{W}} \mathcal{A}$. These two points highlight the adaptivity to contexts and show how causal bound affects the action selection.

Define the following high-probability event

$$\Gamma = \left\{ \forall m \geq 2, \frac{1}{\tau_{m-1}} \sum_{t=1}^{\tau_{m-1}} \mathbb{E}_{w_t, a_t}[(\hat{f}_{m(t)}(w_t, a_t) - f^*(w_t, a_t))^2 | \mathcal{H}_{t-1}] \leq \frac{1}{\rho_m^2} \right\}.$$

The high-probability event and its variants have been proved in literatures (Foster et al., 2018; Simchi-Levi and Xu, 2021; Foster et al., 2020). Our result is slightly different from them as the whole function space is eliminated to $\mathcal{F}^*$. Since these results share the same form, it is straightforward to show $\Gamma$ holds with probability at least $1 - \delta/2$. This is the result of the union bound and the property of the **Least Square Oracle** that is independent of algorithm design.

Our setting do not change the proof procedure of the following lemma (Simchi-Levi and Xu, 2021), because this lemma does not explicitly involve the number of action set. This lemma bounds the prediction error between the true reward and the estimated reward.

**Lemma B.3** *Assume $\Gamma$ holds. For all epochs $m > 1$, all rounds $t$ in epoch $m$, and all policies $\pi \in \Psi$, then*

$$\left| \widehat{\mathcal{R}}_t(\pi) - \mathcal{R}_t(\pi) \right| \leq \frac{1}{2\rho_m} \sqrt{\max_{1 \leq m' \leq m-1} \mathbb{E}_W \left[ \frac{1}{p_{m'}(\pi(W)|W)} \right]}.$$

The third step is to show that the one-step regret $Reg_t(\pi)$ is close to the one-step estimated regret $\widehat{Reg}_t(\pi)$. The following lemma states the result.

**Lemma B.4** *Assume $\Gamma$ holds. Let $c_0 = 5.15$. For all epochs $m$ and all rounds $t$ in epoch $m$, and all policies $\pi \in \Psi$,*

$$Reg(\pi) \leq 2\widehat{Reg}_t(\pi) + c_0\sqrt{\mathbb{E}_W[\mathcal{A}^*(W)]}/\rho_m, \tag{12}$$

$$\widehat{Reg}_t(\pi) \leq 2Reg(\pi) + c_0\sqrt{\mathbb{E}_W[\mathcal{A}^*(W)]}/\rho_m,. \tag{13}$$

*Proof.* We prove this lemma via induction on $m$. It is easy to check

$$Reg(\pi) \leq 1, \widehat{Reg}_t(\pi) \leq 1,$$

as $\gamma_1 = 1$ and $c_0\mathbb{E}_W[\mathcal{A}^*(W)] \geq 1$. Hence, the base case holds.

For the inductive step, fix some epoch $m > 1$ and assume that for all epochs $m' < m$, all rounds $t'$ in epoch $m'$, and all $\pi \in \Psi$, the inequalities (12) and (13) hold. We first show that for all rounds $t$ in epoch $m$ and all $\pi \in \Psi$,

$$Reg(\pi) \leq 2\widehat{Reg}_t(\pi) + c_0\sqrt{\mathbb{E}_W[\mathcal{A}^*(W)]}/\rho_m.$$

We have

$$Reg(\pi) - \widehat{Reg}_t(\pi)$$

$$=[\mathcal{R}(\pi_{f^*}) - \mathcal{R}(\pi)] - [\widehat{\mathcal{R}}_t(\pi_{\hat{f}_m}) - \widehat{\mathcal{R}}_t(\pi)]$$

$$\leq[\mathcal{R}(\pi_{f^*}) - \mathcal{R}(\pi)] - [\widehat{\mathcal{R}}_t(\pi_{f^*}) - \widehat{\mathcal{R}}_t(\pi)]$$

$$\leq|\mathcal{R}(\pi_{f^*}) - \widehat{\mathcal{R}}_t(\pi_{f^*})| + |\mathcal{R}(\pi) - \widehat{\mathcal{R}}_t(\pi)|$$

$$\leq\frac{1}{\rho_m}\sqrt{\max_{1\leq m'\leq m-1}\mathbb{E}_W\left[\frac{1}{p_{m'}(\pi_{f^*}(W)|W)}\right]} + \frac{1}{\rho_m}\sqrt{\max_{1\leq m'\leq m-1}\mathbb{E}_W\left[\frac{1}{p_{m'}(\pi(W)|W)}\right]}$$

$$\leq\frac{\max_{1\leq m'\leq m-1}\mathbb{E}_W\left[\frac{1}{p_{m'}(\pi_{f^*}(W)|W)}\right]}{5\rho_m\sqrt{\mathbb{E}_W[\mathcal{A}^*(W)]}} + \frac{\max_{1\leq m'\leq m-1}\mathbb{E}_W\left[\frac{1}{p_{m'}(\pi(W)|W)}\right]}{5\rho_m\sqrt{\mathbb{E}_W[\mathcal{A}^*(W)]}} + \frac{5\sqrt{\mathbb{E}_W[\mathcal{A}^*(W)]}}{8\rho_m}.$$

The last inequality is by the AM-GM inequality. There exists an epoch $i$ such that

$$\max_{1\leq m'\leq m-1}\mathbb{E}_W\left[\frac{1}{p_{m'}(\pi(W)|W)}\right] = \mathbb{E}_W\left[\frac{1}{p_i(\pi(W)|W)}\right].$$

From Lemma B.2 we know that

$$\mathbb{E}_W\left[\frac{1}{p_i(\pi(W)|W)}\right] \leq \mathbb{E}_W[\mathcal{A}^*(W)] + \mathbb{E}_W[\sqrt{|\mathcal{A}^*(W)|}]\rho_i\widehat{Reg}_t(\pi),$$

holds for all $\pi \in \Psi$, for all epoch $1 \leq i \leq m-1$ and for all rounds $t$ in corresponding epochs.

Hence, for epoch $i$ and all rounds $t$ in this epoch, we have

$$\frac{\max_{1\leq m'\leq m-1}\mathbb{E}_W\left[\frac{1}{p_{m'}(\pi(W)|W)}\right]}{5\rho_m\sqrt{\mathbb{E}_W[\mathcal{A}^*(W)]}}$$

$$=\frac{\mathbb{E}_W\left[\frac{1}{p_i(\pi_{f^*}(W)|W)}\right]}{5\rho_m\sqrt{\mathbb{E}_W[\mathcal{A}^*(W)]}}, \text{ (Lemma B.2:(13))}$$

$$\leq\frac{\mathbb{E}_W[\mathcal{A}^*(W)] + \mathbb{E}_W[\sqrt{|\mathcal{A}^*(W)|}]\rho_i\widehat{Reg}_t(\pi)}{5\sqrt{\mathbb{E}_W[\mathcal{A}^*(W)]}\rho_m}, \text{ (inductive assumption)}$$

$$\leq\frac{\mathbb{E}_W[\mathcal{A}^*(W)] + \mathbb{E}_W[\sqrt{|\mathcal{A}^*(W)|}]\rho_i[2Reg(\pi) + c_0\sqrt{\mathbb{E}_W[\mathcal{A}^*(W)]}/\rho_i]}{5\sqrt{\mathbb{E}_W[\mathcal{A}^*(W)]}\rho_m}, \text{ (Jensen's inequality)}$$

$$\leq\frac{\mathbb{E}_W[\mathcal{A}^*(W)] + \sqrt{\mathbb{E}_W[|\mathcal{A}^*(W)|]}\rho_i[2Reg(\pi) + c_0\sqrt{\mathbb{E}_W[\mathcal{A}^*(W)]}/\rho_i]}{5\sqrt{\mathbb{E}_W[\mathcal{A}^*(W)]}\rho_m}, \text{ } (\rho_i \leq \rho_m \text{ for } i \leq m)$$

$$\leq\frac{2}{5}Reg(\pi) + \frac{1+c_0}{5\rho_m}\sqrt{\mathbb{E}_W[|\mathcal{A}^*(W)|]}.$$

We can bound $\dfrac{\max_{1\leq m'\leq m-1}\mathbb{E}_W\left[\frac{1}{p_{m'}(\pi(W)|W)}\right]}{5\rho_m\sqrt{\mathbb{E}_W[\mathcal{A}^*(W)]}}$ in the same way.

Combing all above inequalities yields

$$Reg(\pi) - \widehat{Reg}_t(\pi) \leq\frac{2(1+c_0)\sqrt{\mathbb{E}_W[\mathcal{A}^*(W)]}}{5\rho_m} + \frac{4}{5}\widehat{Reg}_t(\pi) + \frac{5\sqrt{\mathbb{E}_W[\mathcal{A}^*(W)]}}{8\rho_m}$$

$$\leq\widehat{Reg}_t(\pi) + \left(\frac{2(1+c_0)}{5} + \frac{5}{8}\right)\frac{\sqrt{\mathbb{E}_W[\mathcal{A}^*(W)]}}{\rho_m}$$

$$\leq\widehat{Reg}_t(\pi) + c_0\frac{\sqrt{\mathbb{E}_W[\mathcal{A}^*(W)]}}{\rho_m}.$$

Similarly, we have

$$
\begin{aligned}
&\widehat{Reg}_t(\pi) - Reg(\pi) \\
=&[\widehat{\mathcal{R}}_t(\pi_{\hat{f}_m}) - \widehat{\mathcal{R}}_t(\pi)] - [\mathcal{R}(\pi_{f^*}) - \mathcal{R}(\pi)] \\
\leq&[\widehat{\mathcal{R}}_t(\pi_{\hat{f}_m}) - \widehat{\mathcal{R}}_t(\pi)] - [\mathcal{R}(\pi_{\hat{f}_m}) - \mathcal{R}(\pi)] \\
\leq&|\mathcal{R}(\pi_{\hat{f}_m}) - \widehat{\mathcal{R}}_t(\pi_{\hat{f}_m})| + |\mathcal{R}(\pi) - \widehat{\mathcal{R}}_t(\pi)|.
\end{aligned}
$$

We can bound the above terms in the same steps.

$\square$

*Proof.* Our regret analysis builds on the framework in (Simchi-Levi and Xu, 2021).

**Step 1:** proving an implicit optimization problem for $Q_m$ in Lemma B.2.

**Step 2:** bounding the prediction error between $\widehat{\mathcal{R}}_t(\pi)$ and $\mathcal{R}_t(\pi)$ in Lemma B.3. Then we can show that the one-step regrets $\widehat{Reg}_t(\pi)$ and $Reg(\pi)$ are close to each other.

**Step 3:** bounding the cumulative regret $Reg(T)$.

By Lemma 4 of (Simchi-Levi and Xu, 2021),

$$
\mathbb{E}[Reg(T)] = \sum_{t=1}^{T} \sum_{\pi \in \Psi} Q_{m(t)}(\pi) Reg(\pi).
$$

From Lemma B.4, we know

$$
Reg(\pi) \leq 2\widehat{Reg}_t(\pi) + c_0\sqrt{\mathbb{E}_W[\mathcal{A}^*(W)]}/\rho_m
$$

so

$$
\begin{aligned}
\mathbb{E}[Reg(T)] =& \sum_{t=1}^{T} \sum_{\pi \in \Psi} Q_{m(t)}(\pi) Reg(\pi) \\
\leq& 2\sum_{t=1}^{T} \sum_{\pi \in \Psi} Q_{m(t)}(\pi) \widehat{Reg}_t(\pi) + \sum_{t=1}^{T} c_0\sqrt{\mathbb{E}_W[\mathcal{A}^*(W)]}/\rho_{m(t)} \\
\leq& (2 + c_0)\sqrt{\mathbb{E}_W[\mathcal{A}^*(W)]} \sum_{t=1}^{T} \frac{1}{\rho_{m(t)}} \\
\leq& (2 + c_0)\sqrt{\mathbb{E}_W[\mathcal{A}^*(W)]} \sum_{m=1}^{\lceil \log T \rceil} \sqrt{\log(2\delta^{-1}|\mathcal{F}^*| \log T)\tau_{m-1}/\eta} \\
\leq& (2 + c_0)\sqrt{\mathbb{E}_W[\mathcal{A}^*(W)]} \sum_{m=1}^{\lceil \log T \rceil} \sqrt{\log(2\delta^{-1}|\mathcal{F}^*| \log T)\tau_{m-1}/\eta} \\
\leq& (2 + c_0)\sqrt{\mathbb{E}_W[\mathcal{A}^*(W)] \log(2\delta^{-1}|\mathcal{F}^*| \log T) \sum_{m=1}^{\lceil \log T \rceil} \tau_{m-1}/\eta} \\
\leq& (2 + c_0)\sqrt{\mathbb{E}_W[\mathcal{A}^*(W)] \log(2\delta^{-1}|\mathcal{F}^*| \log T)T/\eta}.
\end{aligned}
$$

$\square$

## B.9    PROOF OF THEOREM 3.2

*Proof.* We first consider $|\mathcal{W}| < \infty$. Since the agent have knowledge about causal bound, any function in $\mathcal{F} - \mathcal{F}^*$ can not be the true reward function. For any given context $w$, the set that the optimal arm will be in is $\mathcal{A}^*(w)$. For any algorithm A, let $A_w$ be the induced algorithm of A when $w$ occurs. Namely, the agent has access to a function space $\mathcal{F}_w = \{f(w, \cdot)|\forall f \in \mathcal{F}^*\}$ and an action set $\mathcal{A}^*(w)$.

From the minimax theorem 5.1 in (Agarwal et al., 2012), we know that there exists a contextual bandit instance such that the regret of $A_w$ is at least $\sqrt{\mathcal{A}^*(w)T_w \log |\mathcal{F}_w|} = \sqrt{\mathcal{A}^*(w)T_w \log |\mathcal{F}^*|}$, where $T_w$ is the number of occurrence of $w$. Hence,

$$Reg(T) \geq \sum_{w \in \mathcal{W}} \sqrt{|\mathcal{A}^*(w)|T_w \log |\mathcal{F}^*|}$$

$$\geq \sqrt{\sum_{w \in \mathcal{W}} |\mathcal{A}^*(w)|T_w \log |\mathcal{F}^*|}.$$

and almost surely,

$$\liminf_{T \to \infty} \frac{Reg(T)}{\sqrt{T}} \geq \sqrt{\sum_{w \in \mathcal{W}} |\mathcal{A}^*(w)| \log |\mathcal{F}^*| \cdot \liminf_{T \to \infty} \frac{T_w}{T}}$$

$$= \sqrt{\sum_{w \in \mathcal{W}} |\mathcal{A}^*(w)| \log |\mathcal{F}^*| \mathbb{P}(W = w)}$$

$$= \sqrt{\mathbb{E}_W[|\mathcal{A}^*(W)|] \log |\mathcal{F}^*|}.$$

Now assume $|\mathcal{W}| = \infty$. Thanks to Glivenko-Cantelli theorem, the empirical distribution converges uniformly to the true reward distribution. We conclude the proof by applying the dominated convergence theorem and the Fubini's theorem, because $\mathcal{A}^*(w)$ is uniformly bounded by $|\mathcal{A}|$.

$\square$

## C    RELATED MATERIALS

**Definition C.1 (Back-Door Criterion)** *Given an ordered pair of variables $(X, Y)$ in a directed acyclic graph $\mathcal{G}$, a set of variables $\mathbf{Z}$ satisfies the back-door criterion relative to $(X, Y)$, if no node in $\mathbf{Z}$ is a descendant of $X$, and $\mathbf{Z}$ blocks every path between $X$ and $Y$ that contains an arrow into $X$.*

**Definition C.2 (d-separation)** *In a causal diagram $\mathcal{G}$, a path $\mathcal{P}$ is blocked by a set of nodes $\mathbf{Z}$ if and only if*

1. *$\mathcal{P}$ contains a chain of nodes $A \leftarrow B \leftarrow C$ or a fork $A \rightarrow B \leftarrow C$ such that the middle node $B$ is in $\mathbf{Z}$ (i.e., $B$ is conditioned on), or*

2. *$\mathcal{P}$ contains a collider $A \leftarrow B \rightarrow C$ such that the collision node $B$ is not in $\mathbf{Z}$, and no descendant of $B$ is in $\mathbf{Z}$.*

*If $\mathbf{Z}$ blocks every path between two nodes $X$ and $Y$, then $X$ and $Y$ are d-separated conditional on $\mathbf{Z}$, and thus are independent conditional on $\mathbf{Z}$.*

If $X$ is a variable in a causal model, its corresponding intervention variable $I_X$ is an exogenous variable with one arrow pointing into $X$. The range of $I_X$ is the same as the range of $X$, with one additional value we can call "off". When $I_X$ is off, the value of $X$ is determined by its other parents in the causal model. When $I_X$ takes any other value, $X$ takes the same value as $I_X$, regardless of the value of $X$'s other parents. If $X$ is a set of variables, then $I_X$ will be the set of corresponding intervention variables. We introduce the following do-calculus rules proposed in (Pearl, 2009).

**Rule 1 (Insertion/deletion of observations)**

$$\mathbb{P}(\mathbf{Y}|do(\mathbf{X}), \mathbf{Z}, \mathbf{W}) = \mathbb{P}(\mathbf{Y}|do(\mathbf{X}), \mathbf{W})$$

if $\mathbf{Y}$ and $I_{\mathbf{Z}}$ are d-separated by $\mathbf{X} \cup \mathbf{W}$ in $\mathcal{G}^*$, the graph obtained from $\mathcal{G}$ by removing all arrows pointing into variables in $\mathbf{X}$.

**Rule 2 (Action/observation exchange)**

$$\mathbb{P}(\mathbf{Y}|do(\mathbf{X}), do(\mathbf{Z}), \mathbf{W}) = \mathbb{P}(\mathbf{Y}|do(\mathbf{X}), \mathbf{Z}, \mathbf{W})$$

if $\mathbf{Y}$ and $I_{\mathbf{Z}}$ are d-separated by $\mathbf{X} \cup \mathbf{Z} \cup \mathbf{W}$ in $\mathcal{G}^{\dagger}$, the graph obtained from $\mathcal{G}$ by removing all arrows pointing into variables in $\mathbf{X}$ and all arrows pointing out of variables in $\mathbf{z}$.

**Rule 3 (Insertion/deletion of actions)**

$$\mathbb{P}(\mathbf{Y}|do(\mathbf{X}), do(\mathbf{Z}), \mathbf{W}) = \mathbb{P}(\mathbf{Y}|do(\mathbf{X}), \mathbf{W})$$

if $\mathbf{Y}$ and $I_{\mathbf{Z}}$ are d-separated by $\mathbf{X} \cup \mathbf{W}$ in $\mathcal{G}^*$, the graph obtained from $\mathcal{G}$ by removing all arrows pointing into variables in $\mathbf{X}$.

# D  CONCLUSIONS

In this paper, we investigate transfer learning in partially observable contextual bandits by converting the problem to identifying or partially identifying causal effects between actions and rewards. We derive causal bounds with the existence of partially observable confounders using our proposed Monte-Carlo algorithms. We formally prove and empirically demonstrate that our causally enhanced algorithms outperform classical bandit algorithms and achieve orders of magnitude faster convergence rates.

There are several future research directions we wish to explore. Firstly, we aim to investigate whether the solutions to discretization optimization converge to those of the original problem. While the approximation error can be exactly zero when all referred random variables are discrete, it is still unclear which conditions for general random variables and discretization methods can lead to convergence. We conjecture that this property may be related to the sensitivity of the non-linear optimization problem.

Lastly, we aim to extend our IGW-based algorithm to continuous action settings. IGW has been successfully applied to continuous action settings and has shown practical advantages in large action spaces. This extension may be related to complexity measures in machine learning.

