# OpenReview forum: "Provably Efficient Learning in Partially Observable Contextual Bandit"
_ICLR.cc/2024/Conference — Submitted to ICLR 2024_

### Official Review · Reviewer_eQpB · 2023-10-29

**Soundness:** 3 good
**Presentation:** 2 fair
**Contribution:** 2 fair
**Rating:** 5
**Confidence:** 3

**Summary:**

In this research, the authors focus on the difficulties associated with transfer learning in situations involving partially observable contextual bandits (TLPOCB), motivated by real-world scenarios like autonomous driving. Traditional bandit algorithms lack prior knowledge and can be computationally intensive. Although transfer learning techniques, which utilize knowledge from related tasks, are applied, they often encounter the problem of biased learned strategies due to incomplete information transfer from experts to agents. To address these issues, the authors introduce novel causal bounds for TLPOCB tasks and develop algorithms that improve learning efficiency.

**Strengths:**

The writing is clear and the problem presented is interesting. The proposed algorithms improve on the existing methods in Li and Pearl (2022).

**Weaknesses:**

The problem setup is spread out into multiple sections and it takes the reader a long time to understand the main topic to be explored. I believe a dedicated section for setup will streamline the reader's understanding.
Some relevant references are missing. For example, there is a line of work including [1],[2],[3] which study the sequential problem under partial observation, characterized by graphs. I think a comparison is needed.

[1] Nicolo Cesa-Bianchi, G'abor Lugosi, Gilles Stoltz, Regret minimization under partial monitoring, Mathematics of Operations Research, 2006

[2] Gábor Bartók, Csaba Szepesvári, Partial Monitoring with Side Information, ALT2012

[3] Tor Lattimore, Csaba Szepesvari, An Information-Theoretic Approach to Minimax Regret in Partial Monitoring, COLT2019

**Questions:**

Since this paper is dealing with a subtle topic at the intersection of bandit, transfer learning, and partial monitoring, a thorough comparison with the existing work should be expected, detailing the connection and difference, which seems missing in the current version.

---

> ### Author Response · Authors · 2023-11-15
>
> Q: The problem setup is spread out into multiple sections and it takes the reader a long time to understand the main topic to be explored.
>
> A: In response to this feedback, we have made improvements to help readers understand the problem setup more efficiently. We have included a notation table in Appendix A, which provides a concise overview of the main notations used in the paper. This addition aims to assist readers in understanding the contributions and key concepts.
>
> Furthermore, we have a preliminaries section that outlines the basic settings related to transfer learning. This section provides foundational information to establish a common understanding before delving into the specifics of the transfer tasks.
>
> To maintain coherence and clarity, we introduce the main contents of each transfer task separately, rather than presenting them all at once. This approach allows readers to grasp the relevant information progressively and maintain a logical flow throughout the paper.
>
> We appreciate the feedback and have taken steps to enhance the organization and presentation of the problem setup in order to make it more accessible to readers.
>
>
>
>
> Q: Since this paper is dealing with a subtle topic at the intersection of bandit, transfer learning, and partial monitoring, a thorough comparison with the existing work should be expected, detailing the connection and difference, which seems missing in the current version
>
>
> A:  We acknowledge the importance of comparing our work with existing literature in order to provide a comprehensive understanding of the field. While it is not feasible to compare every related work within the constraints of a 9-page limit, we have made efforts to provide relevant comparisons in terms of our approaches and theoretical results.
>
> In the paper, we compare our causal bounds with those presented in [1]. We demonstrate the tightness of our bounds by showcasing the limitations of the bounds in [1] and provide a numerical experiment in Section 4 to support our claims.
>
> Additionally, we compare our proposition 3.1 with the result presented in [2] and explain the practical implications of our findings. We also generalize the result in [3] through theorem 3.1 and compare the order dependence on policy space in our theorem 3.3 with [4], highlighting the intuitive improvement achieved.
>
>
> [1] Bounds on causal effects and application to high dimensional data
> [2] Partial counterfactual identification from observational and experimental data.
> [3] Transfer learning in multi-armed bandit: a causal approach.
> [4] Bounding causal effects on continuous outcome.
>
>
>
> Q: Some relevant references are missing
>
> A: In the new version, we cite your mentioned papers in the related work and compare the difference in terms of settings.

---

### Official Review · Reviewer_whdt · 2023-11-01

**Soundness:** 2 fair
**Presentation:** 2 fair
**Contribution:** 2 fair
**Rating:** 3
**Confidence:** 3

**Summary:**

The paper studies PO contextual bandits. In this setting information is transferred between an expert who already has prior information about the joint distribution in this setting, to solve for a setting with unobserved confounders.
- The paper uses linear programming to obtain sample complexity bounds on the regret.
- The paper carries out strong experimental evidence to suggest that the transfer learning algorithm can improve the performance of agents.

**Strengths:**

- The analysis is sound, though notations and clarity can be improved.
- The improvement in regret bound in the given setting is significant. Having said that the advantage to the agent to obtain the improvement in sample complexity is that P(X,Y,W) and separately P(U) is known. The alternative algorithms do not have access to such priors. This expertise from the expert who already knows the joint priors is "transferred" in the transfer learning setting.
- The paper provides a lower bound for their analysis, which is reasonably hard to find.
- The paper shows a significant improvement over baselines in Figure 2.

**Weaknesses:**

- The transfer learning problem for PO Contextual bandits is not well motivated.
- The example on Page 3 is supposed to motivate, "direct approaches can lead to a policy function that is far from the true optimal policy". But I was not able to see this. How do you get the numbers 1.81? Rather, isn't the example motivating the need for considering U in the policy, if available?
- "The expert summarizes its experiences as the joint distribution Fˆ(a, y, w). However, there exists a latent confounder U that affects the causal effects between actions and rewards, making the observed contextual bandit model incomplete. The agent wants to be more efficient and reuse the observed Fˆ(a, y, w) along with the extra prior knowledge about U, i.e., Fˆ(u), to find the optimal policy faster." This is the motivation for the paper.

- The claim on sample complexity of the joint distribution F(a, y , w, u) given F(u) as well as F(a, y ,w) separately, is quite different from learning the full joint distribution from scratch. Therefore the comparison with existing sample complexity analysis is slightly misleading. Specifically, "Our regret demonstrates an improvement in dependence on policy space Π from P(|Π|) to P(log |Π|) compared with ...." is not entirely fair.

- x_ijkl is not very informative as a subscript. Please consider revising this terminology?

---

### Experiments
- The experiments show "that solving non-linear optimization is quite unstable".
- The baselines considered are clear, but the setting is quite unclear. Specifically, is the setting considered general enough, or is it tailored to favor the algorithm proposed in the paper?

**Questions:**

- The authors claim improving regret from sqrt(|P|) to sqrt(log(|P|)), but Table 1 shows regret proportional to sqrt(|A|) for the algorithms proposed.
- In page 3 the authors say, "expert agent can observe the contextual variable in U,W". But also say in Figure 1 caption that "U is the unobserved context". Then why can any expert agent view this?
- Page 4 para 2, what is "skewness from F(u)"?
- If there is an expert who has already learnt the joint distribution F(a, y, w), and further only finite
- In equation 1, can you specify what sets the sup and inf are over?

---

### Minor Suggestions/Typos:
- Table 1: rows 1,3,5 are the classical algorithms?
- Consider changing: "...investigate partially observable contextual bandits, we found few papers focusing on transfer learning in partially observable contextual bandits." --> "...we found few papers focusing on transfer learning in this setting."
- mainly select the arm 1--> mainly selects arm 1.
- Can the notations in Equation 2 be simplified?
- "sequentially solving linear programmings" --> sequentially solve linear programs".

---

> ### Author Response · Authors · 2023-11-15
>
> Q: The transfer learning problem for PO Contextual bandits is not well motivated.
>
> A: Transfer learning in partially observable contextual bandit settings is motivated by practical examples such as autonomous driving. In this scenario, human drivers possess experiential knowledge that is not captured by sensors alone. Works such as [1,2] have explored similar settings and investigated the problem.
>
> Your mentioned "The expert summarizes ... to find the optimal policy faster" also serves as the motivation for this paper.
>
> Q: The example on Page 3 is supposed to motivate, "direct approaches can lead to a policy function that is far from the true optimal policy". But I was not able to see this. How do you get the numbers 1.81? Rather, isn't the example motivating the need for considering U in the policy, if available?
>
>
> A: Thanks for pointing out the typo. We mistakenly added the do notation.
> The intention of the example was to demonstrate that directly transferring the conditional expectation E[Y|A,W] can be suboptimal in certain cases. Of course, having more information about the models will generally be helpful. It is necessary to consider the knowledge of U if it is available, but the way to use U should be carefully designed.
>
>
> Q: The claim on sample complexity of the joint distribution F(a, y , w, u) given F(u) as well as F(a, y ,w) separately, is quite different from learning the full joint distribution from scratch. Therefore the comparison with existing sample complexity analysis is slightly misleading.
>
> A: The knowledge about the distribution only affects the width of the causal bounds. The key difference between existing methods [1,2] and ours lies in transferred online learning algorithms. As we explain, they treat each basis policy as an independent arm in bandit problems, which is often invalid as similar policies can provide information to each other. For example, the optimal policy and the second optimal policy typically yield similar actions and only differ in a small amount of contexts. This is the intuition behind how we improve this order dependence.
>
> To address these challenges, we need to modify the classical IGW technique, which requires non-trivial modifications of its proof. We introduce the subspace of explored policies and show how the causal bounds can shrink its size. You can find the full proof in Appendix
>
> Q: $x_{ijkl}$ is not very informative as a subscript.
>
> A: $x_{ijkl}$ can be understood as representing the PMF of the joint distribution $F(a,y,w,u)$ on the space $\mathcal{A}_i \times \mathcal{Y}_j \times \mathcal{W}_k \times \mathcal{U}_l$.
>
>
> Q: The authors claim improving regret from $\sqrt {|\Pi|}$ to $\sqrt{\log(|\Pi|)}$, but Table 1 shows regret proportional to sqrt(|A|) for the algorithms proposed.
>
> A: $\Pi$ is the policy space and $A$ is the action space. The induced policy space in the function approximation setting is
> $$
> \Pi = \{\pi_f(x): a= \mathop{argmax}_a f(x,a)  }.
> $$
> Since our order dependence on function space $\mathcal{F}$ is $\log |\mathcal{F}^*|$,
> it is shown in Table 1 that we can improve the order dependence from $\sqrt{|\Pi|}$ to $\sqrt{\ln |\Pi|}$,
> compared with existing methods.
>
>
> Q: Question about numerical setup and baselines
>
> A: The numerical setup is explicitly described in the main text, but we provides its details in Appendix A.6. For the first experiment, the values of distributions are randomly generated to provide a general setting. In the second experiment, some values are specifically chosen to demonstrate a significant improvement. This is because in certain cases where there is limited information about the distribution, the improvement achieved by the proposed algorithm may be negligible.
>
> It is important to note that our method does not worsen performance with transferred knowledge. The numerical section of the paper demonstrates that naively transferring knowledge can lead to worse performance for classical algorithms. However, our proposed method does not suffer from negative transfer problems.
>
>
> Q: In page 3 the authors say, "expert agent can observe the contextual variable in U,W". But also say in Figure 1 caption that "U is the unobserved context". Then why can any expert agent view this?
>
> A: U is unobserved to the learner but may be observed by the expert in certain cases.
> Actually, how the learner obtain the required information is not the key factor of our paper.
>
> Q: Page 4 para 2, what is "skewness from F(u)"?
>
> A: It means P(U=0) differs a lot in P(U=1).
>
>
> Q: In equation 1, can you specify what sets the sup and inf are over?
>
> A: sup/inf is taken with respect to all compatible causal models,
> equivalently, all the joint distributions which satisfy the observations.
>
>
> Q: Minor Suggestions/Typos
>
> A: We follow your suggestions and modify our paper accordingly.
>
>
> [1] Transfer learning in multi-armed bandit: a causal approach
> [2] Leaning without knowing: unobserved context in continuous transfer reinforcement learning

---

### Official Review · Reviewer_DQyy · 2023-11-02

**Soundness:** 3 good
**Presentation:** 2 fair
**Contribution:** 3 good
**Rating:** 6
**Confidence:** 2

**Summary:**

This paper considers a transfer learning in a causal bandit problem where a bandit agent receives observational data from an expert agent. In particular, some of the covariates used in the agent is unavailable but partial information is available. Overall, this paper combines the (improved) causal bounds by Li and Pearl and the idea of transfer learning by Zhang and Bareinboim. There are several considerations in improving causal bounds (tigher bounds) and obtaining them pratically (sequential linear programming, estimation error, sampling then optimization). Further, bandit algorithm also has improvement through considering dependency among policies.

**Strengths:**

This paper is more than the combination of transfer learning algorithm (ZB) and causal bounds (LP).
- incorporation of estimation error (in Appendix, which should be in the main paper. It can be only two or three sentences). To make use of small number of available data, naively using maximum likelihood estimates can be misleading. The use of estimation error into bounds is simple yet an important step (especially in transfer learning setting where some of the arms might be truncated) BTW, epsilon in the Algorithm 1 should be highlighted.
- (Eq 3) tigther bounds than Li and Pearl, which is shown to not satisfy constraints over the available information
- practical sampling approach (Eq 4) which avoids rejection sampling.
- The sample, then optimization approach.

**Weaknesses:**

No specific weakness other than the organization, which will be mentioned in the questions (and suggestions) section.

**Questions:**

- Given that ICLR will allow an additional one page, it is expected that the authors will incorporate necessary, important information currently trimmed or in Appendix.
- The paper seems abruptly trimmed or cut during submission. There is no conclusion or discussion. For example, Table 1 should be after Theorem 3.4. Without providing context, it is too abrupt. Further, it seems that table next Table 1 does not have a proper caption. Increase arraystretch to represent table better.
- The organization near Eq 5 and 6 are awkward.
- BTW, lines 1,3,4,6 could be one-liner and Fig 1 can be a lot smaller to afford more space…
- There are many pointers to Appendix which somewhat distracts the flow.
- I suggest Causal Bounds as a separate section and use subsection/paragraph properly. Causal bounds can be described irrelevant to the transfer learning problem. Current paragraphs “Causal Bounds” and “Sampling valid causal models” as a whole can be in the section and can be restructured.
- In Page 8 near at the end, I don’t get the argument here about instrumental variables. If they rely on IV, you may argue that your method is free of IV requirement. It is a bit unnecessary to mention that finding IV is an open problem in academia (Economics?).
- adjust the two plots in Figure 2

By the way, this paper would be more like for AISTATS, CLeaR (causal conference), NeurIPS, ICML ... I wonder why the authors pick the ICLR as the 'best' revenue to present the results. I don't see any part 'representation learning' involved. Given the type of the audience, I rate this paper 'marginally above' the threshold. Otherwise, I will raise to 'accept'.

---

> ### Author Response · Authors · 2023-11-15
>
> Q: Suggestions on structures.
>
>
> A: Thank you for your feedback. Following your suggestions, we utilize the additional one page allowed by ICLR to incorporate the necessary information,
> including a conclusion subsection and brief descriptions about appendix structure.
> The following content is added in the additional one page.
>
> "
> In this paper, we investigate transfer learning in partially observable contextual bandits by converting the problem to identifying or partially identifying causal effects between actions and rewards.
> We derive causal bounds with the existence of partially observable confounders using our proposed Monte-Carlo algorithms.
> We formally prove and empirically demonstrate that our causally enhanced algorithms outperform classical bandit algorithms and achieve orders of magnitude faster convergence rates.
>
> There are several future research directions we wish to explore.
> Firstly, we aim to investigate whether the solutions to discretization optimization converge to those of the original problem.
> While the approximation error can be exactly zero when all referred random variables are discrete,
> it is still unclear which conditions for general random variables and discretization methods can lead to convergence.
> We conjecture that this property may be related to the sensitivity of the non-linear optimization problem.
>
> Lastly, we aim to extend our IGW-based algorithm to continuous action settings.
> IGW has been successfully applied to continuous action settings and has shown practical advantages in large action spaces.
> This extension may be related to complexity measures in machine learning.
> "
>
>
> "
> In appendix A, we put the omitted information regarding the examples, algorithms, theorems, implementation details and numerical setup.
> We also put partial related work section due to the strict page limitation of ICLR.
> In appendix B, you can find all proofs about claimed theorems.
> In appendix B, you can find some related materials about causal tools.
> In appendix D, we generalize our sampling method on general simplex more than the special one defined by the special optimization (1).
> In appendix E, we put a conclusion section and discuss the future work.
> "
>
>
> For other suggestions, we modify our paper accordingly according to the page limit and modification requirement.
>
>
>
>
>
> Q: There are many pointers to Appendix which somewhat distracts the flow.
>
>
> A:  Thank you for your feedback. We understand your concern regarding the pointers to the appendix and their potential impact on the flow of the paper. We have included these pointers to ensure that important details and results are not omitted due to the strict page limitation. By providing references to the relevant sections in the appendix, we aim to maintain the logical flow of the paper while still providing comprehensive information. We will make sure that the pointers are clear and effectively guide readers to the corresponding sections in the appendix.
>
>
>
>
>
>
> Q: In Page 8 near at the end, I don't get the argument here about instrumental variables. If they rely on IV, you may argue that your method is free of IV requirement.
> It is a bit unnecessary to mention that finding IV is an open problem in academia (Economics?).
>
>
> A: We modify this sentence in the updated version.
> We actually want to show that our method does not rely on IVs as you mentioned in your reviews.
>
>
>
> Q: this paper would be more like for AISTATS, CLeaR (causal conference), NeurIPS, ICML ... I wonder why the authors pick the ICLR as the 'best' revenue to present the results. I don't see any part 'representation learning' involved.
>
> A: We appreciate your perspective on the suitability of the paper for different conferences. While ICLR is commonly associated with representation learning, it also welcomes submissions from various topics, including reinforcement learning, causal learning, and transfer learning. Our paper falls within the scope of ICLR as it addresses transfer learning in the context of causal reinforcement learning. We acknowledge that representation learning may not be explicitly mentioned in our paper, but the broader topics covered in ICLR encompass the themes and techniques we explore. Moreover, ICLR has accepted papers in the areas of reinforcement learning, causal learning, and transfer learning in the past.

---

> > ### Comment · Reviewer_DQyy · 2023-11-21
> > **Follow-up**
> >
> > Thank you for providing detailed responses and incorporating my inquiries effectively. I have carefully read the reviews you provided. However, there are a few aspects that haven’t been addressed. I would like to ask about those.
> > Firstly, I suggested having a separate section for causal bounds and an overall restructuring, but it seems it hasn’t been addressed. I’m curious about the authors’ opinions on this matter. Additionally, I have some additional questions about the organization of equations (Eq5 and 6) and the position of Table 1. Lastly, in Fig.2, the sizes of the x-axis annotations do not match each other. I kindly request adjustments to be made in this regard.

---

> > > ### Author Response · Authors · 2023-11-21
> > >
> > > Q: a separate section for causal bounds and an overall restructuring
> > >
> > > A: Firstly, making significant changes to the structure may not be allowed by the ICLR guidelines,
> > > as the final version of the PDF should not differ substantially from the original submitted version.
> > > Secondly, Subsection 3.1 already provides a comprehensive description of computing causal bounds,
> > > as the content about MAB in this subsection is limited.
> > > To enhance coherence and maintain a clear focus on the topic of transfer learning,
> > > we have to add some information about bandits at the beginning and end of Subsection 3.2.
> > > This will help establish the connection between causal bounds and transfer learning tasks,
> > > as the objectives of causal bounds are different vary across transfer learning tasks.
> > > Since computing causal bounds is an intermediate step in our method,
> > > we include it as a subsection within the section "Transfer Learning via Causal Inference."
> > >
> > > Our remaining concern raised from your review is whether the title "transfer learning in MAB" of Subsection 3.1 is suitable.
> > > If you have more suggestions, we are glad to accept.
> > >
> > >
> > > Q: the organization of equations, the position of Table 1, Fig.2
> > >
> > > A: We have made adjustments to improve the organization.
> > > Specifically, we have reordered the content for Equations 5 and 6.
> > > In the current version, a paragraph titled "Estimation Error" has been added at the end of this subsection.
> > > Additionally, we have placed Table 1 at the end of Section 3.
> > > As for Fig. 2, we have adjusted the figures accordingly.

---

> > > > ### Comment · Reviewer_DQyy · 2023-11-22
> > > >
> > > > Thanks for the response. I was initially thinking about separating a causal bound section or renaming some of the sections but now it seems clear that there is no need to do it. (The phrase 'overall restructuring' was in fact splitting the section. Sorry for the confusion.)

---

### Official Review · Reviewer_yo3f · 2023-11-09

**Soundness:** 2 fair
**Presentation:** 1 poor
**Contribution:** 2 fair
**Rating:** 3
**Confidence:** 3

**Summary:**

The paper studies partially observable contextual bandits, where the context comprises 2 variables W, U and the agent have access to W only while U is hidden. The problem is mapped to the framework of causal effects between actions and rewards and formulated as an optimization problem that is solved using sampling and LP techniques. The proposed algorithms are shown to have orders of magnitude better regret than classical bandit algorithms.

**Strengths:**

- The considered problem is important and has multiple applications.
- The approach is novel.
- The resulting regrets are smaller than those of existing bandit algorithms.

**Weaknesses:**

- The writing of the paper can be substantially improved. Many of the definitions and arguments are not clear to me from a theoretical aspect. Please see my questions below regarding this.

- The regret definition is written in terms of W only (not U). At each time slot, do you compete with a policy that has access to the true function $f^*$ and both realizations of W, U or you compete against policy that has access to $f^*$, W and take expectation over U?

- What is h(a) in table (1)?

- What is sup/inf in equation (1)? Is it either inf or sup, both will work? Do you mean that one will give an upper bound and the other a lower bound? This should be clearly stated. In the same paragraph you mention that (1) gives a bound on the causal effects. Causal effects between which variables? What is the mathematical formula for the causal effect to see that (1) gives an upper bound?

- If the algorithms are applied in the famous case of contextual linear bandits, what would be the resulting regret in that case?

- The works in [1,2,3] consider contextual linear bandits with known context distribution without observing the realization. Even though the setup is more limited, I believe the authors need to provide a comparison when the results of the paper are limited to the setups of [1,2,3].

The paper has a novel idea, but it is hard to follow the math and verify the results. I suggest the authors make the paper more clear and rigorous. Please explain the results with more math and logic. A table of notations would also help. I will update my score after reading the authors response.

[1] Kirschner, Johannes, and Andreas Krause. "Stochastic bandits with context distributions." Advances in Neural Information Processing Systems 32 (2019).
[2] Hanna, Osama, Lin Yang, and Christina Fragouli. "Learning from Distributed Users in Contextual Linear Bandits Without Sharing the Context." Advances in Neural Information Processing Systems 35 (2022): 11049-11062.
[3] Hanna, Osama A., Lin Yang, and Christina Fragouli. "Contexts can be cheap: Solving stochastic contextual bandits with linear bandit algorithms." The Thirty Sixth Annual Conference on Learning Theory. PMLR, 2023.

**Questions:**

Please see. weaknesses.

---

> ### Author Response · Authors · 2023-11-15
>
> Q: The regret definition is written in terms of W only (not U). At each time slot, do you compete with a policy that has access to the true function and both realizations of W, U, or do you compete against a policy that has access to $f^*$, W, and take the expectation over U?
>
> A: Since the learner cannot observe $U$, we compete against a policy that has access to $f^*$, W, and takes the expectation over U. The realizability assumption only needs to hold in the sense of taking expectation with respect to $U$.
>
>
> Q: What is h(a) in Table (1)?
>
> A: h(a) is the causal upper bound solved by the optimization problem (1). Please refer to the MAB part in section 3.1 for more details.
>
>
> Q: What is sup/inf in equation (1)? Is it either inf or sup, and both will work? Do you mean that one will give an upper bound and the other a lower bound? This should be clearly stated. In the same paragraph you mention that (1) gives a bound on the causal effects. Causal effects between which variables? What is the mathematical formula for the causal effect to see that (1) gives an upper bound?
>
>
> A: Equation (1) means we can get upper and lower bounds of its object by solving corresponding maximization and minimization problems, respectively.
> The object is the causal effect between action $A$ and reward $Y$ and both its upper and lower bounds are needed in the transfer learning algorithms.
> Theorem A.1 in Appendix shows the mathematical expression of the object
> and in the last sentence on page 4, we show the expression of the object in the discrete setting.
>
>
> Q: If the algorithms are applied in the famous case of contextual linear bandits, what would be the resulting regret in that case?
>
>
> A: We have provided the implementation of infinite function spaces in section A.5. It is worth noting that our algorithm and theorems can be easily extended to handle infinite $\mathcal{F}$ using standard learning-theoretic tools such as metric entropy. Let's assume $\mathcal{F}$ is equipped with a maximum norm $||\cdot||$. We can consider an $\epsilon$-covering $\mathcal{F}^*_{\epsilon}$ of $\mathcal{F}^*$ under the maximum norm. Since $|\mathcal{F}^*_{\epsilon}|$ is finite, we can directly replace $\mathcal{F}^*$ with $\mathcal{F}^*_{\epsilon}$ without changing any algorithmic procedures. Thanks to the property of $\epsilon$-covering, there exists a function $f_{\epsilon}^*\in \mathcal{F}^{\epsilon}$ such that $|| f_{\epsilon}^* - f^* || \leq \epsilon$. Hence, the regret can be bounded by the expression:
>
> $$
> Reg(T) \leq 8 \sqrt{ \mathbb{E}_W[ \mathcal{A}(W) ] T \log (2\delta^{-1}  |\mathcal{F}^*\_{\epsilon}|   \log T ) } + \epsilon T.
> $$
>
> By replacing the dependence on $\log |\mathcal{F}^*|$ in the algorithm's parameters with $\log|\mathcal{F}^*_{\epsilon}|$ and setting $\epsilon=\frac{1}{T}$, we obtain a similar result, up to an additive constant of 1.
>
> In the context of linear spaces, the quantity $N(\mathcal{F}^*,||\cdot||,\epsilon)$, which represents the covering number, scales with $\sqrt{\log N(\mathcal{F}^,||\cdot|| ,\epsilon)}$. For linear spaces, such a quantity is of order $O(d\log(\frac{1}{\epsilon}))$.
> Hence, the improvement of regret order will be $N(\mathcal{F},||\cdot||,\epsilon) - N(\mathcal{F}^*,||\cdot||,\epsilon)$ in terms of function spaces.
>
> It is important to note that $N(\mathcal{F}^*,||\cdot||,\epsilon) \leq N(\mathcal{F},||\cdot||,\epsilon)$ since $\mathcal{F}^* \subset \mathcal{F}$. The covering number clearly demonstrates how extra causal bounds help improve the algorithm's performance by shrinking the search space. The transfer learning algorithm only needs to search within a spherical shell with a thickness of at most $M-m$, where $m = \inf_{w,a} l(w,a)$ and $M=\sup_{w,a} h(w,a)$. The bounds chip away the surface of the unit sphere and scoop out the concentric sphere of radius $m$.
>
>
>
> Q: The works in [1,2,3] consider contextual linear bandits with known context distribution without observing the realization. Even though the setup is more limited, I believe the authors need to provide a comparison when the results of the paper are limited to the setups of [1,2,3].
>
> A: Thank you for your suggestion. We have added the mentioned papers to the related work section. Please refer to the updated version of our paper for more details.

---

### Author Response · Authors · 2023-11-15

We sincerely appreciate all of the reviewers for providing valuable suggestions and feedback on our paper. We have carefully considered their input and have made several updates to improve the quality of our work. Here is a summary of the changes we have made:

1. We have addressed all the typos and mistakes pointed out by the reviewers.

2. In response to the feedback, we have added a notation subsection in the appendix and included a linker in Table 1. These additions aim to assist readers in quickly reviewing and better understanding our work.

3. To enhance the comprehensiveness of our paper, we have incorporated more related work as suggested by reviewers.

4. Although ICLR's policy limits extensive changes, we have managed to improve the coherence of our paper by adding new sentences that align with the requirements.

5. To provide further clarity and structure to our work, we have included an additional page illustrating the organization and content of our appendix.

In order to emphasize the novelty of our results, we would like to briefly highlight some key points that may have been overlooked by readers:


1. By proposing a practical sampling approach, we have achieved tighter causal bounds compared to existing methods. We have supported our findings with both theoretical explanations and numerical experiments.
Moreover, as mentioned by Reviewer DQyy,
incorporation of estimation error into bounds is simple yet an important step.



2. In the task of function approximation,  we have introduced a novel idea to improve the order dependence on policy space, surpassing existing approaches. Our intuition stems from the notion that treating each basis policy as an independent arm in bandit problems is not ideal. Instead, similar policies can provide valuable information to one another. We have introduced a new adaptivity technique to enhance the classical IGW technique and have demonstrated how the size of the explored policy subspace can affect the causal bounds.

3. We are pleased to acknowledge that most reviewers have recognized the significant improvement in regret bounds within the given setting. Both in theory and numerical experiments, our work has demonstrated notable advancements. In particular, Reviewer whdt has noted that "the paper provides a lower bound for their analysis, which is reasonably hard to find.


Once again, we express our gratitude to all the reviewers for their valuable feedback, which has greatly contributed to the enhancement of our paper.

---

### Meta-Review · Area_Chair_fCdE · 2023-12-06

**Metareview:**

The paper considers transfer learning in partially observable contextual bandits. The problem is mapped to identifying or partially identifying causal effects between actions and rewards and formulated as an optimization problem that is solved using sampling and LP techniques. The new sampling approach enjoyed a tighter causal bound. The proposed causal bandit algorithm achieves improved regret compared to classic bandit algorithm.

The reviewers appreciate the novel approach and the improvement in regret bounds by leveraging causal effects. However, there are shared concerns that the writing and presentation needs improvement, the motivation is not clear, and comparison to partial monitoring literature is missing. The rebuttal answered some questions but did not fully address the concerns. The authors are suggested to revise the paper according to the comments. Considering the significance of the improved bounds, the revised paper would be a strong submission in next venue.

**Justification For Why Not Higher Score:**

Three out of four reviewers recommend rejection. The AC agrees with this decision. There are shared concerns that the writing and presentation needs improvement, the motivation is not clear, and comparison to partial monitoring literature is missing. The rebuttal answered some questions but did not fully address the concerns.

**Justification For Why Not Lower Score:**

N/A

---

### Decision · Program_Chairs · 2024-01-16

Reject